



# Mechanical and hydraulic properties of the excavation damaged zone (EDZ) in the Opalinus Clay of the Mont Terri Rock Laboratory, Switzerland

Sina Hale[1], Xavier Ries[1], David Jaeggi[2], Philipp Blum[1]

[1]Karlsruhe Institute of Technology (KIT), Institute of Applied Geosciences (AGW), Kaiserstr. 12, 76131 Karlsruhe, Germany
[2] Federal Office of Topography (swisstopo), Seftigenstr. 264, 3084 Wabern, Switzerland

*Correspondence to*: Sina Hale (sina.hale@kit.edu)

**Abstract.** Construction of cavities in the subsurface is always accompanied by excavation damage. Especially in the context of deep geological nuclear waste disposal, the evolving excavation damaged zone (EDZ) in the near field of emplacement
tunnels is of utmost importance concerning safety aspects. As the EDZ differs from the intact host rock due to enhanced hydraulic transmissivity and altered geomechanical behavior, reasonable and location-dependent input data on hydraulic and mechanical properties is crucial. Thus in this study, a hydro-mechanical characterization of an EDZ in the Mont Terri underground rock laboratory, Switzerland, was performed using three different handheld devices: (1) air permeameter, (2) microscopic camera and (3) needle penetration test. The discrete fracture network (DFN), consisting of artificially induced
unloading joints and reactivated natural discontinuities, was investigated by a portable air permeameter as well as combined microscopic imaging with automatic evaluation. Geomechanical and geophysical characterization of the claystone was conducted based on needle penetrometer testing at the exposed rock surface. Within the EDZ, permeable fractures with a mean hydraulic aperture of $84 \pm 23$ µm are present. Under open conditions, self-sealing of fractures is suppressed and cyclic long-term fracture aperture oscillations in combination with closure resulting from convergence processes, is observed. Based on
measured needle penetration indices, a uniaxial compressive strength of $30 \pm 13$ MPa (normal to bedding) and $18 \pm 8$ MPa (parallel to bedding) was determined. Enhanced strength and stiffness is directly related to near-surface desaturation of the claystone and a sharp decrease in water content from 6.6 wt.-% to 3.7 wt.-%. The presented methodological approach is particularly suitable for time-dependent monitoring of EDZs since measurements are nondestructive and do not change the actual state of the rock mass. This allows for a spatially resolved investigation of hydraulic and mechanical fracture apertures,
fracture surface roughness as well as physico-mechanical rock parameters and their intra-facies variability.

## 1 Introduction

For all types of man-made underground structures, formation of a so-called excavation damaged zone (EDZ) or excavation disturbed zone (EdZ) is inevitable (Pusch and Stanfors, 1992; Shen and Barton, 1997). As geologic formations are affected by regional or local stress fields, stress redistribution during excavation leads to displacement and convergence, accompanied by





the formation of unloading fractures in the rock mass around the cavity (Bossart et al., 2002). The EDZ is characterized by

severe hydraulic, mechanical and geochemical modifications as well as newly formed connected porosity (Dao et al., 2015;

Kupferschmied et al., 2015; Labiouse and Vietor, 2014; Sato et al., 2000; Yong et al., 2017). Thus, significant changes in flow

and transport properties can be observed in the EDZ due to an enhanced permeability of the connected fracture network serving

as preferential flow paths. In the EdZ, flow and transport properties are only scarcely affected as potentially saturated fractures

are mainly isolated and occurring processes are reversible (Bossart et al., 2002, 2004; Tsang et al., 2005).

The EDZ and its impact on hydraulic and mechanical rock properties are of particular importance for the underground storage

of radioactive material (Blümling et al., 2007; Fairhurst, 2004). According to the current state of knowledge, multi-barrier

systems for geological disposal are the preferred option for effectively isolating high-level nuclear waste and spent fuels

(Birkholzer et al., 2012; Chapman and Hooper, 2012). This repository concept is designed for a service life of up to one million

years, which will essentially be guaranteed by the sealing function of a natural barrier (Apted and Ahn, 2010; Wilson and

Berryman, 2010). In Switzerland, the Opalinus Clay, an overconsolidated Jurassic claystone, was selected as a host rock for

deep geological storage of high-level radioactive waste (Bossart et al., 2017; Nagra, 2002). In the context of host rock

characterization and site assessment, the generic underground rock laboratory (URL) Mont Terri provides a valuable site for

research, testing and development of in-depth technical know-how. Since 1996, numerous studies and experiments have been

conducted in order to evaluate the hydrogeological, mechanical, geochemical and transport properties of the undisturbed and

altered rock, and to examine the behavior of the Opalinus Clay when exposed to short- or long-term THMC impacts (Bossart

et al., 2017; Pearson et al., 2003). Besides the Mont Terri URL in Switzerland, a number of underground laboratories in other

countries and their potential or selected host rock formations are in operation, mainly in crystalline rocks (e.g. Äspö Hard Rock

Laboratory in Sweden) or sedimentary rocks including plastic clays (e.g. HADES URL in Belgium) and stiff or indurated clays

(e.g. Meuse/Haute-Marne URL and Tournemire URL in France) (Blechschmidt and Vomvoris, 2010; Delay et al., 2014).

Similar to the Opalinus Clay in Switzerland, the EDZ and its impact on hydro-mechanical characteristics of the rock mass in

the near field of underground structures is of particular interest for the Callovo-Oxfordian claystone in France (e.g. Armand et

al., 2014; Baechler et al., 2011; Menaceur et al., 2016) as well as for the Boom Clay in Belgium (e.g. Bastiaens et al., 2007;

Dao et al., 2015).

In the Opalinus Clay of the Mont Terri URL, the EDZ is characterized by a significantly enhanced hydraulic conductivity of

$1 \times 10^{-14}$ to $1 \times 10^{-5}$ m s$^{-1}$ (Bossart et al., 2004; Jaeggi and Bossart, 2014; Marschall et al., 2017), whereas for undisturbed

conditions it ranges between $2 \times 10^{-14}$ to $5 \times 10^{-12}$ m s$^{-1}$ (Jaeggi and Bossart, 2014; Lavanchy and Mettier, 2012). In intact rock,

substances are only transported by diffusive processes (Croisé et al., 2004). Within the EDZ, advective transport is facilitated

due to fracture permeability, which is several orders of magnitude higher than the matrix permeability of the claystone

(Marschall et al., 2017). Hydraulic fracture parameters such as permeability, transmissivity or flow rate are in turn directly

related to the hydraulic fracture aperture $a_h$ (Zimmerman and Bodvarsson, 1996), which therefore represents a key parameter

for assessing the hydraulic characteristics of a fractured rock mass or an EDZ. The hydraulic aperture is usually derived from

the cubic law (Louis, 1969; Snow, 1965) and relates to the mean opening width of a fracture accessible to advective transport.



Due to the confirmed self-sealing capacity of the Opalinus Clay caused by swelling of mixed-layer illite-smectite clay minerals
(e.g. Bernier et al., 2007), the hydraulic conductivity of the EDZ is expected to decline within a period of several tens to hundreds of years by progressive fracture closure (Jaeggi and Bossart, 2014). In addition, fractured rock masses are also characterized by a pronounced hydro-mechanical coupling, i.e. changes in the mechanical stress state result in changes of permeability and therefore hydraulic fracture aperture (Cammarata et al., 2007; Min et al., 2004; Rutqvist and Stephansson, 2003). Generally, $a_h$ is nonlinearly linked to the mechanical fracture aperture $a_m$ in dependence of fracture surface roughness
(Blum et al., 2009; Renshaw, 1995), for example via the Barton–Bandis model using the well-known Joint Roughness Coefficient (JRC) (Barton, 1982; Barton et al., 1985). The mechanical fracture aperture represents the average geometrical distance between the fracture surfaces (e.g. Hakami and Larsson, 1996) and is needed to examine the response of fracture networks due to normal or shear stresses, such as fracture dilation or closure mechanisms (e.g. Blümling et al., 2007; Cuss et al., 2011; Zhang, 2016) or mechanical self-sealing of artificial fractures (e.g. Marschall et al., 2017; Nagra, 2002).

Similar to the hydraulic properties, mechanical properties of the Opalinus Clay diverge significantly depending on direction, facies and stress regime (Bock, 2009; Giger et al., 2015). Furthermore, a strong dependency of failure strength and elastic parameters on water content or suction was proven (Jaeggi and Bossart, 2014; Wild et al., 2015). Due to this marked and clay-specific hydro-mechanical coupling (Amann et al., 2017; Marschall et al., 2017), geomechanical parameters such as uniaxial compressive strength, tensile strength and shear strength as well as the Young´s modulus of the Opalinus Clay generally
increase with decreasing water content (Blümling et al., 2007; Wild et al., 2015). In the EDZ in Mont Terri, elastic P- and S-wave velocity is significantly reduced due to the high degree of damage (Schuster et al., 2001, 2017). Furthermore, geomechanical properties of the Opalinus Clay in the EDZ are also modified in comparison to the undisturbed rock mass. In the short-term, a reduction in effective stress caused by pore pressure excess in the vicinity of the advancing excavation front leads to early damage of the rock around the cavity. Right after excavation, pore water drainage, pore pressure dissipation as
well as increased suction of the rock mass can be observed close to the cavity (Amann et al., 2017; Giger et al., 2015). In the long-term, a general decrease in water content caused by dehydration of the rock leads to locally enhanced rock strength and stiffness (Wild et al., 2015).

An accurate and comprehensive hydraulic and mechanical characterization of the EDZ is therefore essential for confirming the integrity of the host rock in terms of risk and performance assessment (e.g. Blum et al., 2005; Popp et al., 2008; Tsang et
al., 2015; Xue et al., 2018). This key information serves as an appropriate starting point for numerical modeling studies investigating the development of the EDZ in the post-closure phase of the repository, and is also useful for the selection and adaptation of engineering designs or adequate constructional measures (e.g. Hudson et al., 2005; Marschall et al., 2017; Nagra, 2019; Tsang et al., 2012). This does not only apply to the issue of nuclear waste disposal, but also generally to other underground structures in different geological materials and settings (e.g. Li et al., 2012; Sheng et al., 2002; Wu et al., 2009).

Hydraulic fracture apertures are usually determined in the laboratory by permeameter tests, with either gases or liquids being used to flow through fractured rock samples (Kling et al., 2016; Li et al., 2018; Shu et al., 2019; Zhang, 2018). In the field, hydraulic properties can be derived from hydraulic or pneumatic borehole tests (Aoyagi and Ishii, 2019; Jakubick and Franz,





1993; de La Vaissière et al., 2015; Shao et al., 2008). For hydraulic field testing, many experimental designs are practicable that were already used in the Mont Terri URL, for example, hydraulic pulse, constant head or constant rate withdrawal tests using single-, double- or multi-packer systems (Bossart et al., 2004; Martin et al., 2004; Meier et al., 2000; Yu et al., 2017). Mechanical fracture apertures can generally be obtained by different fracture imaging methods, whereby visibility can be improved by injecting dyed or fluorescent resin into the fractured rock (Armand et al., 2014; Bossart et al., 2002).

Seismic velocity measurements can be carried out in the laboratory by using ultrasonic pulse devices (Popp et al., 2008; Wild et al., 2015) and in the field by applying mini-seismic methods (Schuster et al., 2017) or by conducting interval velocity logs in boreholes (Martin et al., 2004; Schuster et al., 2001). Geomechanical strength and deformation parameters are usually determined by laboratory experiments. For this purpose, many different test setups are utilized such as compressive strength tests, tensile strength tests, shear tests and triaxial tests under drained or undrained conditions. For the Opalinus Clay, numerous geomechanical tests were carried out on drill cores, primarily examining bedding anisotropy as well as the hydro-mechanical coupling by adapting the water content of the samples (Amann et al., 2011, 2012, 2017; Wild et al., 2015). In the field, handheld probes such as Schmidt hammer or needle penetrometer are used to estimate the uniaxial compressive strength and other mechanical parameters of rock material (Aydin, 2009; Buyuksagis and Goktan, 2007; Erguler and Ulusay, 2009; Ulusay and Erguler, 2012).

For most investigations drilling is required, either directly for performing borehole tests or for taking standard-compliant samples. However, drilling is not always feasible, for example due to poor accessibility and because it is very time consuming and expensive. Furthermore, boreholes also affect the EDZ by creating additional fluid pathways. Core samples do not necessarily reflect the initial state as they can suffer from disturbance or damage during extraction and transport, leading for example to a change in water content. Hence, the objective of this study is to investigate the hydro-mechanical properties of the EDZ in the Opalinus Clay of the Mont Terri URL using on-site measurements on the exposed rock surface. In this study, a nondestructive determination of hydraulic and mechanical parameters is conducted by using a transient-flow air permeameter, a microscope camera and a needle penetration test. The presented methodological approach provides valuable information about the influence of the EDZ on the hydraulic and mechanical properties of the Opalinus Clay. Furthermore, alterations within the EDZ of a non-lined niche due to direct air exposure of the rock surface are investigated and discussed.

## 2 Material and methods

### 2.1 Study site

Fieldwork was performed in the EZ-B niche of the Mont Terri underground rock laboratory (URL) in St. Ursanne, Switzerland (Fig. 1a). The axis of the niche is oriented almost normal to Gallery 04 and to the minimum principal stress direction of the in situ stress field (Yong et al., 2010). The niche is located in the upper shaly facies of the Opalinus Clay, which consists of dark gray, mostly mica and pyrite containing calcareous silty-sandy claystones (Hostettler et al., 2017). Bedding is dipping 45 ° towards 150 °, thus the niche axis is oriented perpendicular to the strike of the bedding. As the URL is located in the southern





limb of the Mont Terri overthrust anticline, the Opalinus Clay has experienced tectonic deformation (Nussbaum et al., 2011). As a consequence, pre-existing natural discontinuities, i.e. bedding-parallel tectonic faults, steeper splays and bedding planes, are present (Nussbaum et al., 2005). The EZ-B niche provides direct access to the overconsolidated claystone of the shaly facies of the Opalinus Clay and to the excavation-induced fracture network of Gallery 04. It was excavated from December 2004 to March 2005 by mainly road header and pneumatic hammering (Nussbaum et al., 2005). Numerous experiments were

carried out in the niche, focusing for example on determining the extent and degree of damage of the EDZ (Schuster et al., 2017), fracture network analysis and small-scale mapping (Nussbaum et al., 2011; Yong, 2007), or long-term hydro-mechanical coupling processes (Möri et al., 2010; Ziefle et al., 2017).

Excavation-induced unloading joints (EDZ fractures) that are related to the construction of Gallery 04 are present within the first 1.3 m of the EZ-B niche (Nussbaum et al., 2005). Strike direction is mostly parallel to Gallery 04 and therefore

perpendicular to the axis of the niche. At greater distances, artificial EDZ fractures that originate from the excavation process of the EZ-B niche itself are mainly oriented parallel to the sidewalls. In addition to the artificially induced unloading fractures, the EDZ also includes tectonic faults and splays, referred to as tectonic fractures. These tectonic fractures were reactivated by stress redistribution and convergence processes after the niche excavation and therefore show measurable fracture apertures (Nussbaum et al., 2005, 2011).  In contrast, tectonic discontinuities outside the EDZ are completely closed. In the entrance

area of the EZ-B niche, the rock is partly covered by shotcrete, making a section of the EDZ inaccessible (Fig. 1b). The on-site measurements in the Mont Terri URL were carried out on 16–17 April 2019. At that time, the average air temperature in the EZ-B niche was 16.5 °C, while relative humidity was in the range of 67–72 %.



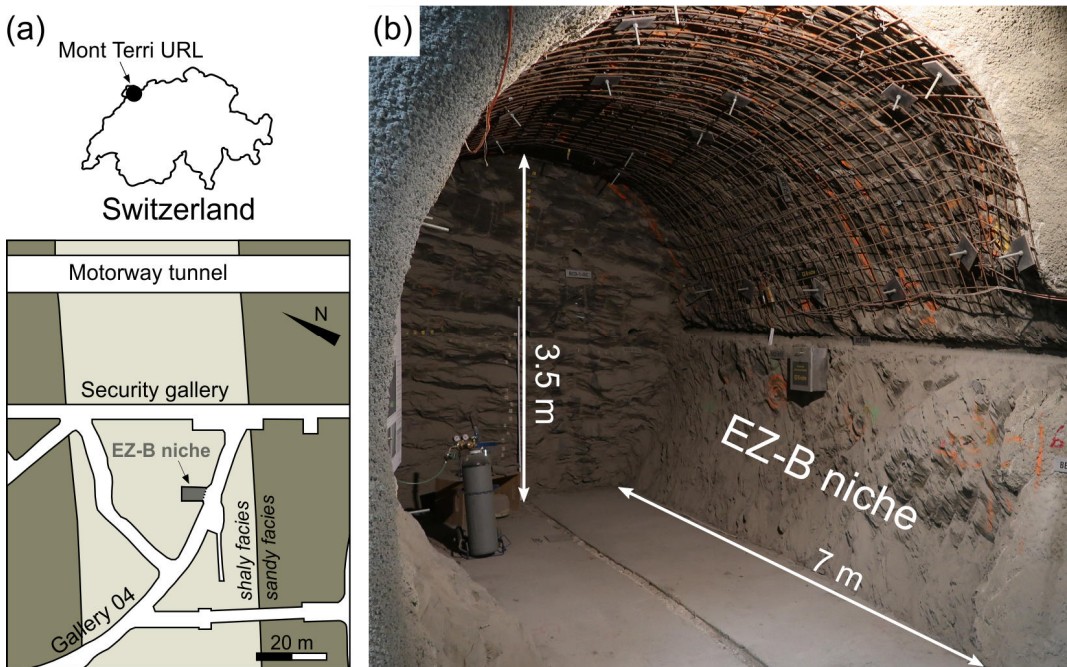

**Figure 1: (a) Location of the Mont Terri Underground Rock Laboratory (URL) alongside the security gallery of the Mont Terri**
**motorway tunnel and the EZ-B niche situated in the shaly facies of the Opalinus Clay. (b) Photo and dimensions of the EZ-B niche,**
**where data acquisition was conducted. In the entrance area, shotcrete partly covers the rock surface on the left and right side wall.**

**2.2 Air permeameter**

A handheld transient-flow air permeameter (model TinyPerm 3, New England Research Inc.) was used to measure the
hydraulic aperture ($a_h$) of accessible fractures in the EZ-B niche. The working principle of the device was outlined by Brown
and Smith (2013) and illustrated in Fig. 1 of Hale et al. (2020b). Further specifications are provided by New England Research,
Inc. (2015). For each fracture, measurement was repeated at least three times. In case the mean absolute deviation of measured
values was above 10 µm, the measurement was continued. Clear outliers were rejected in order to eliminate erroneous data,
e.g. caused by fracture fillings (dust or loose material) or by leaks at the rubber nozzle tip of the air permeameter. Hydraulic
fracture apertures are determined directly based on the time-dependent pressure equilibration and the internal calibration of
the device (Brown and Smith, 2013; New England Research, Inc., 2015). Thus, no post-processing of data is required for the
air permeameter.

For most rocks, the hydraulic aperture derived from air permeameter measurements agrees with the hydraulic aperture available
for advective flow. For sandstone, this was demonstrated by Cheng et al. (2020), where the air permeameter was validated by
steady-state flow tests and different types of artificial fractures with apertures ranging between 7 and 62 µm. For all tested
samples, hydraulic apertures were in excellent agreement with deviations below 5 µm (Cheng et al., 2020). Since clay minerals
represent the main constituents of the Opalinus Clay (Bossart and Thury, 2008), diffusive double layers (DDL) are formed on
exposed clay mineral surfaces in water saturated fractures (Soler, 2001), which could potentially reduce the hydraulic aperture





of fractures in argillaceous rocks. For the Opalinus Clay, the maximum thickness of the DDL is only 22 nm, which can be approximated by the Debye length (Wigger and Van Loon, 2018) using representative pore water ionic strength values (e.g.
Pearson et al., 2003; Van Loon et al., 2003). Thus, in this case the DDL effect on $a_h$ is negligible.

## 2.3 Microscope camera

For the same set of fractures (Sect. 2.2), high-resolution images of fracture traces were taken with a microscope camera (DigiMicro Mobile, dnt GmbH) in order to estimate mechanical fracture apertures ($a_m$) in the EZ-B niche. The digital camera, with an image resolution of up to 12 million pixels, comprises of a microscope with an adjustable magnification factor of up
to 240. By adjusting the focus dial, the rock surface can be brought into sharp focus. Subsequently, the set magnification factor has to be recorded to evaluate the images. While taking the photo, the field of view should be aligned parallel to the fracture axis and the camera should look vertically into the fracture.

Microscope camera images can be evaluated both manually and automatically. The arithmetic mean of distances measured evenly along the fracture trace corresponds to the mechanical fracture aperture $a_m$, whereas the associated standard deviation
($\sigma_{a_m}$) provides a reasonable measure for fracture surface roughness (e.g. Brown, 1987; Kling et al., 2017). The manual evaluation method uses an image analysis software to determine the distance between the two fracture edges regularly along the imaged segment. For detailed description of the manual image analysis approach, we refer to Hale et al. (2020b). A minimum number of 20 distance measurements was needed to gain representative mechanical apertures. Additionally, an automatic approach for determining $a_m$ and $\sigma_{a_m}$ was applied in this study. The code for running the workflow in Fig. 2 is
written in MATLAB (see Data and Code availability). As input data, microscopic grayscale images with specified magnification factors are used. For an applied image resolution of 9 million pixels, the resulting image size is 3456 pixels in x-direction. As the images should be cropped adequately before analysis according to the extent of the fracture void area, the image size in y-direction is variable (Fig. 2).





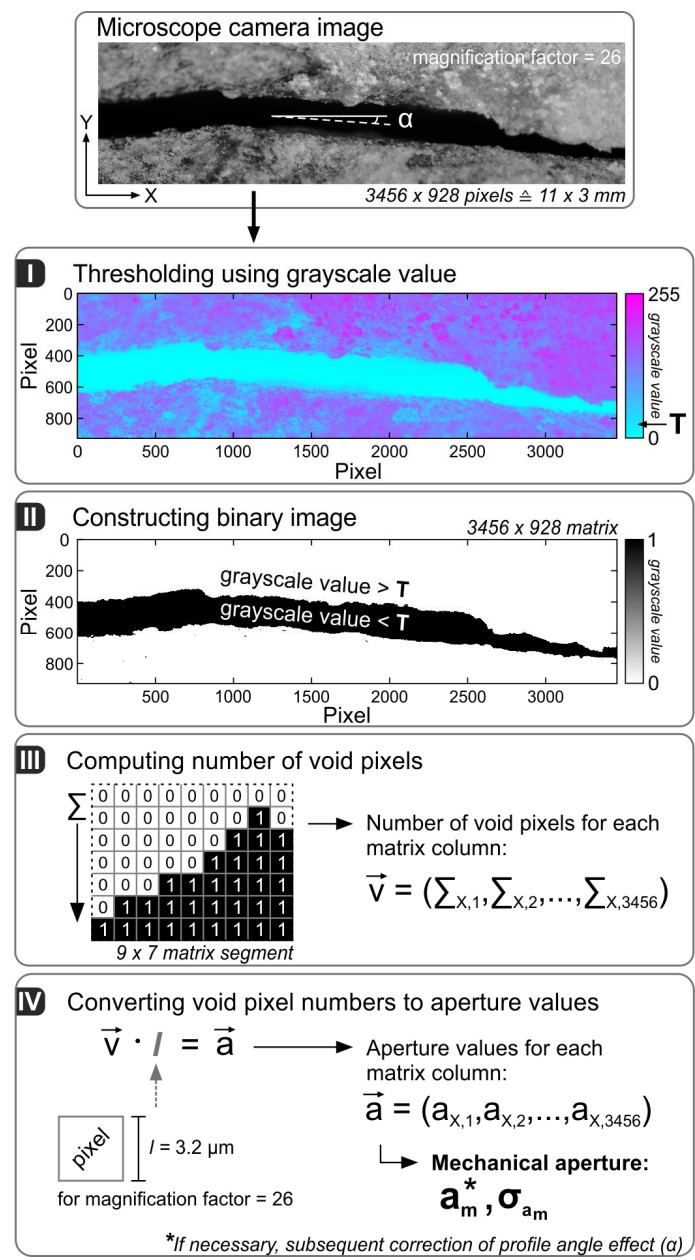

**Figure 2: Workflow of the automatic approach for determining the mechanical fracture aperture based on microscope camera images of fracture trace segments.**

The automated workflow is delineated in Fig. 2 and involves four steps. Based on the grayscale values of the image (0–255), a suitable threshold value $T$ is first defined in order to segment the void area (region of interest) as precisely as possible. As a second step, the image is binarized, i.e. a value of 1 is assigned to the pixels of the void area and a value of 0 is assigned to all remaining pixels. Based on the resulting binary pixel matrix, the total number of void pixels ($\vec{v}$) is determined column-wise. In order to convert the number of void pixels into aperture values $\vec{a}$ (in µm), the real length of one image pixel is required as





a conversion factor. The pixel length directly depends on the magnification of the microscope camera that was set when taking the photo. It can be determined by using the software PortableCapture (Hale et al., 2020b). Finally, the mechanical aperture of the analyzed fracture segment corresponds to the arithmetic mean of the computed aperture values in $\vec{a}$. If the fracture trace

deviates from the x-direction of the image (denoted by angle $\alpha$), $a_m$ is corrected accordingly. Using $a_m$ and $\sigma_{a_m}$, hydraulic fracture apertures $a_h$ can be estimated subsequently by applying different empirical equations (Table 1).

**Table 1: Equations for estimating the hydraulic fracture aperture based on the mean mechanical fracture aperture and the standard deviation of measured distance values along a fracture trace.**

| Reference | Equation | |
|---|---|---|
| Kling et al. (2017) | $a_h \approx a_m \left(1 + \dfrac{\sigma_{a_m}}{a_m}\right)^{-1.5}$ | (1) |
| Rasouli and Hosseinian (2011) | $a_h \approx a_m \sqrt[3]{abs\left(1 - 2.25\dfrac{\sigma_{a_m}}{a_m}\right)}$ | (2) |
| Barton and de Quadros (1997) | $a_h \approx a_m \dfrac{1}{\sqrt[3]{1 + 20.5\left(\dfrac{\sigma_{a_m}}{2a_m}\right)^{1.5}}}$ | (3) |
| Xiong et al. (2011) | $a_h \approx a_m \sqrt[3]{abs\left(1 - \dfrac{\sigma_{a_m}}{a_m}\right)}$ | (4) |
| Renshaw (1995) | $a_h \approx (\overline{a_m})_{Geom}$ | (5) |
| Matsuki et al. (1999) | $a_h \approx \sqrt[3]{1 - \dfrac{1.13}{1 + 0.191\left(\dfrac{2a_m}{\sigma_{a_m}}\right)^{1.93}}}$ | (6) |
| Renshaw (1995) | $a_h \approx a_m \dfrac{1}{\sqrt[3]{\left(1 + \dfrac{\sigma_{a_m}^2}{a_m^2}\right)^{1.5}}}$ | (7) |
| Amadei and Illangasekare (1994) | $a_h \approx a_m \dfrac{1}{\sqrt[3]{1 + 0.6\left(\dfrac{a_m}{\sigma_{a_m}}\right)^{-1.2}}}$ | (8) |





**205**   **2.4 Needle penetration test**

A needle penetrometer device (Model SH-70, Maruto Corporation Limited, Japan) was used to determine the needle penetration index ($NPI$) of the Opalinus Clay normal and parallel to bedding, which is directly dependent on the strength of the rock (e.g. Ulusay and Erguler, 2012). For testing, the needle is pushed into the rock by manually applying a maximum load of 100 N. The quotient of the applied load and the attained needle penetration depth (in N mm$^{-1}$) corresponds to the $NPI$

**210**   (Aydan et al., 2014; Ulusay et al., 2014). Needle penetrometer tests were carried out at different measurement points on the rock surfaces in the EZ-B niche. If microcrack formation around the needle hole or tensile splitting along bedding planes was observed, the measured value was excluded from the dataset. For detailed description of working principle and testing procedure we refer to the ISRM suggested method for needle penetration test by Ulusay et al. (2014).

Several physico-mechanical parameters are directly related to $NPI$. For example, a strong correlation between $NPI$ and

**215**   uniaxial compressive strength of intact rock was proved (e.g. Aydan, 2012; Uchida et al., 2004). Established empirical equations were used in this study to estimate uniaxial compressive strength ($UCS$), Brazilian tensile strength ($BTS$), Young´s modulus ($E$), elastic P-wave ($v_p$) and S-wave velocity ($v_s$), cohesion ($c$) and friction angle ($\varphi$) (Table 2). In order to enable a direct comparison of the estimated parameters with existing literature data, the water content of the Opalinus Clay was additionally determined by oven drying according to DIN EN ISO 17892-1 (2015-03-00) using two representative rock

**220**   specimens from the walls of the EZ-B niche, sampled at the time of the on-site measurements.



Table 2: Equations for the estimation of physico-mechanical rock parameters using the needle penetration index ($NPI$), taken from Ulusay and Erguler (2012), Ulusay et al. (2014), and Aydan et al. (2014). The relations are based on compiled experimental data obtained from various lithologies.

| Parameter | Equation | Reference |
|---|---|---|
| Uniaxial compressive strength [MPa] | $UCS \approx 0.418 \cdot NPI - 0.004$ | Uchida et al. (2004) [a] |
| | $UCS \approx 0.2 \cdot NPI$ | Aydan (2012) [b] |
| | $UCS \approx 0.402 \cdot NPI^{0.929}$ | Ulusay and Erguler (2012) [c] |
| Brazilian tensile strength [MPa] | $BTS \approx 0.02 \cdot NPI$ | Aydan et al. (2014) [b] |
| Young's modulus [GPa] | $E \approx 0.05 \cdot NPI$ | Aydan et al. (2014) [b] |
| Cohesion [MPa] | $c \approx 0.04 \cdot NPI$ | Aydan et al. (2014) [b] |
| Friction angle [°] | $\varphi \approx 13.375 \cdot NPI^{0.25}$ | Aydan et al. (2014) [b] |
| P-wave velocity [km s$^{-1}$] | $v_p \approx 0.33 + 0.3 \cdot NPI^{0.5}$ | Aydan et al. (2014) [b] |
| S-wave velocity [km s$^{-1}$] | $v_s \approx 0.1 + 0.18 \cdot NPI^{0.5}$ | Aydan et al. (2014) [b] |

[a] clay
[b] mudstone, sandstone, siltstone, marl, lignite, tuff, soapstone, pumice, soft limestone, sheared shale
[c] marl, siltstone, mudstone, tuff

In Ulusay and Erguler (2012), the term needle penetration resistance ($NPR$) is used instead of $NPI$.

# 3 Results and discussion

## 3.1 Hydraulic and mechanical fracture properties

### 3.1.1 Measured hydraulic fracture aperture

The hydraulic aperture $a_h$ of artificially induced unloading joints, reactivated fault planes and bedding-parallel desiccation or unloading cracks of the EDZ in the EZ-B niche of the Mont Terri URL was determined at 43 measuring points on both side walls using the handheld transient-flow air permeameter (Fig. 3a). The mean hydraulic fracture aperture in the EZ-B niche was $84 \pm 23\,\mu m$, with values in the range of around $100\,\mu m$ occurring most frequently (Fig. 3b). On average, artificially induced unloading fractures (hereinafter referred to as EDZ fractures), mainly oriented sub-parallel to the axis of Gallery 04, showed the smallest hydraulic apertures of $61 \pm 30\,\mu m$ compared to reactivated fault and bedding planes. They were also





characterized by the largest range of measured aperture values from 20 to 100 µm, which is also evident from the high standard deviation.

Reactivated tectonic discontinuities, namely fault planes and splays of the SSE-dipping thrust system (hereinafter referred to as tectonic fractures), showed an average hydraulic aperture of 89 ± 18 µm. The hydraulic fracture aperture of bedding-parallel cracks was highest (94 ± 8 µm), although the obtained average value cannot be considered representative due to a small number

of measurements. On the right side wall of the EZ-B niche, most of the sampling points were arranged near borehole BEZ-B1 (Fig. 3b) due to good accessibility and beneficial surface conditions. Based on the measured aperture values, no indication of a "borehole damaged zone" (Amann et al., 2017) is observable, which is related to the fact that the borehole BEZ-B1 is oriented perpendicular to the strike of the bedding.



Figure 3: (a) Structural maps of the EZ-B niche with measurement points for hydraulic and mechanical aperture determination on the left (closed symbols) and right side wall (open symbols), modified after Nussbaum et al. (2005). (b) Hydraulic fracture apertures measured by air permeameter plotted against distance to Gallery 04. On the right side, the distribution of $a_h$ is visualized by a probability density function obtained by Kernel density estimation (KDE).





The on-site aperture measurements in the EDZ clearly show that about 15 years after excavation, hydraulically open fractures
and therefore accessible fluid pathways are still present in the EZ-B niche. In the Opalinus Clay, fractures are successively
closed by self-sealing processes which lead to a significant permeability reduction in the EDZ, finally approaching the
hydraulic conditions of undeformed rock again (Bernier et al., 2007; Jaeggi and Bossart, 2014; Nagra, 2002). However, this
only applies for saturated conditions after backfilling and sealing, when progressive resaturation of the host rock around
underground facility is initiated (Bossart et al., 2017; Marschall et al., 2017). Under open conditions the EDZ is an unsaturated
zone, as for example shown by Ziefle et al. (2017). On-site measurements and numerical simulations confirm the evolution of
a seasonally influenced desaturated zone with an extent of about 50 cm as well as a long-term influenced desaturated zone
with an extent of up to 2.2 m after 15 years (Ziefle et al., 2017). In the unsaturated EDZ in non-lined niches, seasonal changes
of humidity and temperature inhibit self-sealing processes. Since 2006, the temporal evolution of the EDZ in the EZ-B niche
was assessed by jointmeter time series obtained from a single tectonic fracture. Based on this dataset, a cyclic long-term closure
of the monitored fracture was demonstrated (Ziefle et al., 2017), but this is probably primarily due to niche convergence.
For shales or argillaceous rocks, changes in the saturation state are directly linked to structural modifications (Valès et al.,
2004; Yurikov et al., 2019). In the highly saturated state, a large portion of the pore water is adsorbed onto the clay mineral
surfaces, while for high external suction, i.e. desaturation, it is extracted from the rock through the pore network (Zhang et al.,
2007). Dehydration of claystone leads to a substantial modification of the pore size distribution and to a decrease in total
porosity (Yurikov et al., 2019). It also induces a reduction of pore and swelling pressure, which in turn impedes self-sealing
processes (Tsang et al., 2005; Zhang et al., 2007). In addition, secondary shrinking-induced tension fractures can develop
parallel to bedding with progressive dehydration, especially if the exposed claystone surface was subject to multiple drying–
wetting cycles (Delage, 2014). In contrast, hydration of claystones leads to an increase in total porosity and pore pressure as
well as to a significantly enhanced creep and swelling capacity, but it can also be accompanied by mineral precipitation within
the fracture network (Yurikov et al., 2019; Zhang et al., 2007). In case of the non-lined EZ-B niche in Mont Terri, self-sealing
is inhibited due to a sharp decrease in water content of the rock mass close to the niche caused by sustained ventilation since
tunnel excavation. This observation is of particular importance for the second phase in repository development, the open drift
stage, where ventilation-induced damage and dehydration in the tunnel systems is also expected (Tsang et al., 2005). The
presented results therefore serve as a valuable analogue and provide information on the state of the EDZ in a non-lined niche
in indurated clay after prolonged exposure.
With increasing distance to Gallery 04, a general decrease in hydraulic fracture aperture $a_h$ was expected. When leaving the
EDZ, the degree of damage or disturbance generally decreases as deconfinement, displacement and deviatoric stresses within
the rock mass are highest directly next to the cavity (Lisjak et al., 2016; Yong et al., 2010). Based on the results of the air
permeameter measurements, however, a weak positive correlation between $a_h$ and the horizontal distance could be observed
(correlation coefficient $r = 0.43$). A decrease in hydraulic aperture with greater distance to Gallery 04 could not be observed,
since the fractures in the EZ-B niche originate from two different excavations. Due to the applied excavation technique and
favorable orientation of the niche (Sect. 2.1), the EDZ of the EZ-B niche is less pronounced compared to the EDZ around





Gallery 04. However, two EDZs, i.e. two fracture systems, are superposed. Hydraulic apertures in the immediate vicinity of Gallery 04 are comparatively small (Fig. 3b). Presumably, this is caused by shotcrete application to the exposed claystone
surface associated with increased water availability, leading to partial re-saturation of the rock. This water supply most likely promoted swelling of clay minerals and fracture closure to a certain degree, resulting in a general reduction of hydraulic apertures in the entrance area of the EZ-B niche.

Due to the limited measuring range of the air permeameter, hydraulic apertures of widely opened fractures ($a_h$ > 2 mm) could not be quantified. Thus, the mean $a_h$ of open fractures in the EZ-B niche was rather underestimated with this method. It is also
noticeable that for 49 % of the measurement points, the first measured value was smallest. This can be explained by dust or loose material inside the fracture which was removed by the first stroke with the air permeameter. In most cases, the first value was therefore excluded from the data according to Sect. 2.2. As outlined above and supported by continuously recorded jointmeter data from the EZ-B niche, the saturation state or moisture content of the Opalinus Clay has a major influence on measured fracture apertures. From 2015 to 2019, the investigated single fracture was subject to annual aperture fluctuations of
up to 500 μm. This phenomenon is mainly related to seasonal fluctuations of relative humidity ($RH$) (Ziefle et al., 2017). However, over the long period since the niche was completed in 2005, the observed trend of cyclic aperture closure has decreased substantially and seasonal fluctuations of fracture aperture are by now far less pronounced. According to the long-term monitoring of climatic conditions in the EZ-B niche, relative humidity is highest between July and October ($RH$ = 100 %), while the lowest values are usually recorded in February ($RH \approx 60$ %). Since the measurement campaign for the present study
was carried out in April, it can be assumed that the obtained hydraulic and mechanical aperture values roughly represent an annual average state of the continuously changing fracture system.

For the Opalinus Clay in the Mont Terri URL, no direct information on hydraulic fracture apertures is available, which further illustrates the difficulty of conducting practicable and accurate $a_h$ measurements in the field. In order to compare the measured values with literature values in terms of plausibility, fracture transmissivities ($T_f$) obtained from extensive hydraulic testing in
the Mont Terri URL were utilized (Table 3). Based on the cubic law, an equivalent hydraulic fracture aperture ($a_{h,eq}$) can be calculated from $T_f$ using the relation (Brown, 1987)

$$a_{h,eq} = \sqrt[3]{T_f \cdot \frac{12 \cdot \mu_W}{\rho_W \cdot g}} \qquad (9)$$

where $\mu_W$ is kinematic viscosity of water, $\rho_W$ is density of water and $g$ is the gravitational acceleration. It should be noted that hydraulic tests are generally used to characterize certain borehole intervals. Transmissivity values that are derived from these
tests therefore relate to a certain rock volume, while the number of hydraulically active fractures intersecting the test interval is usually unknown (Gustafson and Fransson, 2006). In this case, fracture densities must be considered for calculating equivalent hydraulic apertures. However, some studies also provide single fracture transmissivities which could therefore directly be converted to $a_{h,eq}$ using Eq. (9) for direct comparison with $a_h$ measured by the air permeameter (Table 3).



**Table 3: Comparison of measured hydraulic fracture aperture (this study) with equivalent hydraulic aperture values derived from reported single fracture transmissivities from the Opalinus Clay in the Mont Terri URL.**

| Reference | Description | Fracture transmissivity | Equivalent hydraulic aperture |
|---|---|---|---|
| Martin et al. (2004) | Transmissivity from high-resolution probe testing in four boreholes (EDZ + undisturbed rock) | $T_f = 1.0 \times 10^{-14} - 1.0 \times 10^{-9}$ m² s⁻¹ | < 1–11 µm [a] |
| Meier et al. (2000) | Single fracture transmissivity from long-term hydraulic testing (EDZ) | $T_f = 2.0 \times 10^{-8}$ m² s⁻¹ | 29 µm |
| Bossart et al. (2004) | Single fracture transmissivity from hydraulic cross-hole testing (EDZ) | $T_f = 1.9 \times 10^{-8} - 4.0 \times 10^{-8}$ m² s⁻¹ | 29–37 µm |
| Blümling et al. (2007) | Reported maximum local fracture transmissivity (EDZ) | $T_f = 5.0 \times 10^{-7}$ m² s⁻¹ | 85 µm |
| This study | Air permeameter (EDZ) | – | 20–112 µm |

[a] assumption of a single hydraulically dominant fracture (short test intervals, typically 5 cm)

Although slightly larger, the hydraulic fracture apertures measured by air permeameter are in the same order of magnitude compared to equivalent hydraulic apertures that were derived from single-fracture transmissivity testing (Table 3). Due to the previously outlined processes related to the successive dehydration of the claystone, hydraulic apertures in the EZ-B niche were expected to differ slightly from the literature values due to the elongated exposure time. Nevertheless, the measured values are plausible and clearly show that the air permeameter is suitable for the measurement of hydraulic fracture apertures within the EDZ of the Opalinus Clay.

### 3.1.2 Measured mechanical fracture aperture

For the same set of fractures, the mechanical fracture aperture $a_m$ was determined based on microscope camera images. For seven measurement points the mechanical aperture could not be evaluated due to poor quality of the microscopic images. The measured mechanical fracture apertures in the EZ-B niche showed a widespread range between 19 µm and 833 µm, while most values were clustered around 115 µm (Fig. 4). Artificial EDZ fractures showed the lowest values among the studied fracture types (127 ± 92 µm), whereas the mean mechanical aperture in the EZ-B niche was 233 ± 205 µm. Again, no distinct trend with increasing distance to Gallery 04 could be observed. For the vast majority of sampled fractures, $a_m$ was greater than or equal to $a_h$ as expected (Fig. 4). As is the case for hydraulic apertures, almost no information on mechanical fracture apertures within the EDZ of the Opalinus Clay was available for comparison. Bossart et al. (2004) mention unloading fractures



with mechanical apertures of up to 1 cm. However, as the specified measurement range of the air permeameter is limited to hydraulic apertures of 2 mm, such widely opened fractures were not investigated in this study.

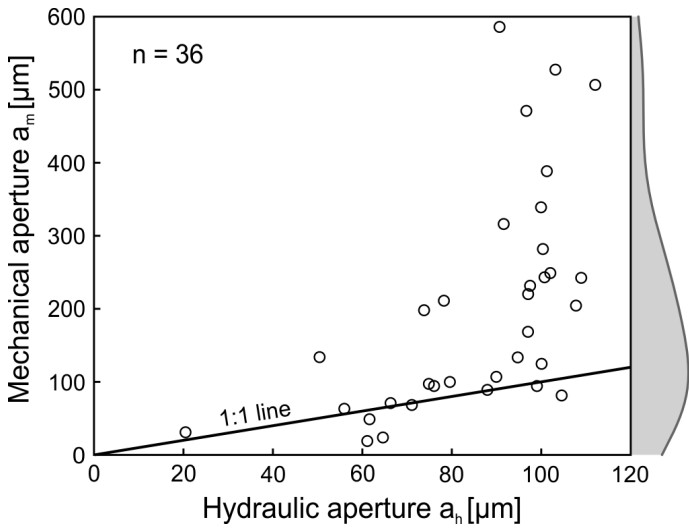


**Figure 4: Comparison of measured mechanical and hydraulic fracture apertures in the EZ-B niche with probability density function of $a_m$. Two outliers with mechanical apertures of 789 µm and 833 µm are not shown for better visibility.**

The newly implemented analysis algorithm for microscopic images (Fig. 2) provided similar results compared to the manual approach used in Hale et al. (2020b) (correlation coefficient $r = 0.93$). Thus, automatic image analysis is highly recommended.

In addition to significant time savings, more representative results for $a_m$ and $\sigma_{a_m}$ of the imaged fracture trace segment can be obtained due to the large number of distance measurements. For images that could not be evaluated automatically, manual analysis was employed. This particularly applied to images that were not sufficiently focused or to fissures with very small mechanical apertures. In these cases, the void area could not be properly distinguished from the rock, thus hampering the selection of an appropriate threshold value for automatic analysis.

### 3.1.3 Estimated hydraulic fracture aperture


Based on the mean mechanical aperture ($a_m$) and the corresponding standard deviation ($\sigma_{a_m}$), providing a statistical measure for fracture surface roughness, hydraulic apertures were estimated using established empirical relations (Eq. (1) to (8), Table 1). For eight different equations, the results are shown in Fig. 5. With regard to the mean hydraulic fracture aperture, the full dataset of the discrete fracture network (DFN) in the EZ-B niche was best reproduced by Eq. (1) (Kling et al., 2017). The

median of this data set was also in excellent agreement with the measured data (Fig. 5). However, there were generally large deviations between the measured and estimated mean $a_h$. With regard to the frequency distribution of data points, Eq. (6) to (8) performed better because $a_m$ is reduced to a lesser extent. Similar to the dataset of the TinyPerm 3 (air permeameter), most values were clustered around 100 µm. However, few very high values led to a significant overestimation of the arithmetic





mean (216–221 µm). In order to provide a representative measure of central tendency of the estimated hydraulic aperture for
a given fracture set or DFN, the use of the median is highly recommended. For EDZ fractures in particular, $a_h$ was best
estimated by Eq. (2) (Rasouli and Hosseinian, 2011).

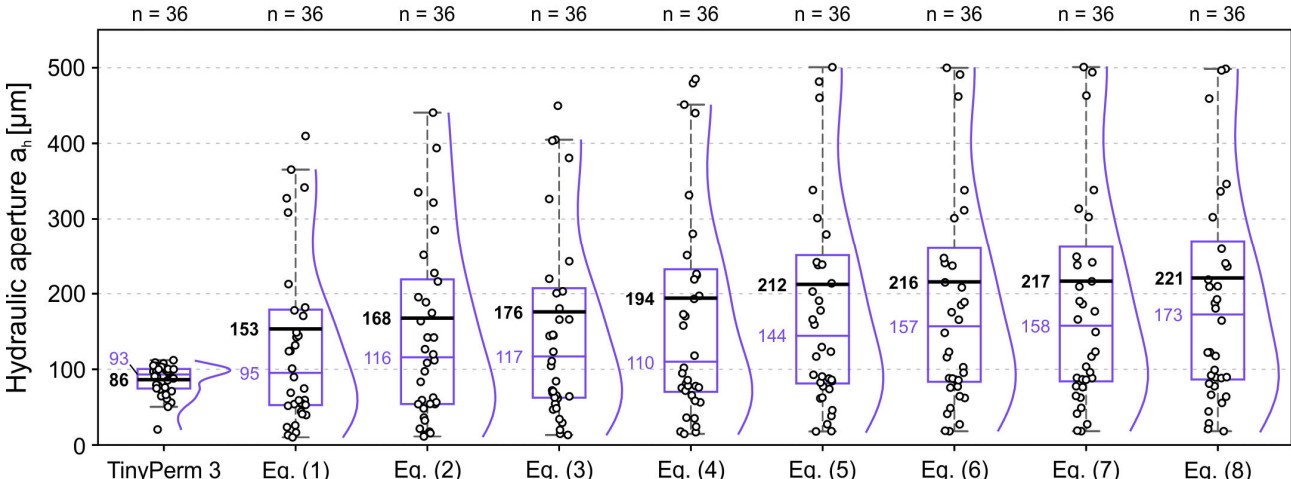

**Figure 5: Comparison of hydraulic fracture apertures measured by air permeameter (TinyPerm 3) and estimated based on mean**
**mechanical fracture apertures and standard deviations from microscopic fracture trace analysis. For each dataset, the arithmetic**
**mean (bold black line), the median (thin purple line) and the probability density function (obtained by Kernel density estimation) is**
**shown. Values above 550 µm are excluded from the graph.**

Higher deviations between estimated and measured hydraulic apertures were observed for fractures with either very low or
very high mechanical apertures in relation to the measured value range. This was confirmed by a strong correlation between
$a_m$ and the resulting deviation between estimated and measured $a_h$ values, for example, for Eq. (1) the correlation coefficient
was 0.96. It was noticeable that for EDZ fractures a far higher agreement of estimated and measured hydraulic apertures could
be observed in comparison to tectonic fractures, independently from the relation that was used for conversion. This is most
likely related to smaller mechanical apertures. While EDZ fractures seem to better correspond to the general model concept of
an "ideal plane-parallel fracture", tectonic fractures are characterized by a different $a_m$–$a_h$–relation.

Despite the good performance of the microscope camera method for EDZ fracture analysis, direct measurement of $a_h$ with the
air permeameter is always preferable. Nevertheless, the data derived from microscopic imaging can provide explicit
information on fracture geometry and formation mechanism. Presumably due to their formation mechanism, EDZ fractures
(artificial unloading joints) showed comparatively little variation in aperture along the fracture traces, resulting in a rather low
mean standard deviation $\sigma_{a_m}$ of 58 µm. For EDZ fractures, $a_m$ was on average 1.7 times larger than $a_h$. In contrast, for
tectonic fractures (reactivated fault planes), an average $\sigma_{a_m}$ of 90 µm and a $a_m/a_h$ ratio of 2.4 was determined in the EZ-B
niche. However, the higher mean standard deviation for the measured aperture values along the fracture segment obtained for
tectonic fractures is not necessarily caused by a higher roughness of the fault surfaces themselves. Namely, observations in the
Mont Terri URL showed that fault surfaces are generally polished with slickensides, while EDZ fracture surfaces often show





plumose structures (Nussbaum et al., 2005, 2011; Yong, 2007). On the other hand, since EDZ fractures originate from tensile

stresses, these fracture surfaces also show a high degree of matching. This directly leads to a comparatively small standard deviation of measured aperture values along an imaged EDZ fracture segment. Fault planes in the EZ-B niche were reactivated by stress redistribution during excavation and subsequent convergence, which is why the investigated tectonic fractures even show measurable apertures. Due to shear loading, relative displacement led to a higher mismatch of the two fracture surfaces. For the tectonic fractures, the higher mean standard deviation $\sigma_{a_m}$ that was observed by microscopic image analysis therefore

primarily reflects the increased variance of measured distances due to mismatched fracture walls rather than the actual fracture surface roughness.

## 3.2 Geomechanical and geophysical properties

Needle penetration testing (NPT) was performed at 47 sampling points on both side walls of the EZ-B niche, including 18 tests normal to bedding and 29 tests parallel to bedding. The measured needle penetration index ($NPI$) clearly confirmed the

significant strength anisotropy of the Opalinus Clay (Jaeggi and Bossart, 2014). Normal to bedding, strength was significantly higher than parallel to bedding, indicated by a $NPI$ of $98 \pm 29$ N mm$^{-1}$ and $59 \pm 19$ N mm$^{-1}$, respectively. For all measurements a constant load of 100 N was selected. Thus, the $NPI$ was only depending on the observed needle penetration depth. Measurements normal to bedding resulted in several invalid tests due to formation of microcracks around the needle hole. In this case, the measured value was discarded (Sect. 2.4). The results of the parameter estimation based on the measured $NPI$

are listed in Table 4.

Figure 6 provides a graphical overview and enables comparison with available literature data to assess the quality of the applied in situ parameter estimation. As discussed in Sect. 1, geomechanical and geophysical parameters in the EZ-B niche are significantly affected by the desaturation of the rock. Compared to the natural water content of the shaly facies of about 6.6 wt.-% (Bossart and Thury, 2008), the water content of the Opalinus Clay in the EZ-B niche has decreased drastically to 3.7 wt.-%

due to direct air contact and long-time ventilation of the URL (Jaeggi and Bossart, 2014; Ziefle et al., 2017). Since samples for water content determination were taken directly from the rock surface at the walls of the EZ-B niche, this value represents the state of the Opalinus Clay after 15 years of direct atmospheric exposure. The $NPI$, and therefore estimated geomechanical and geophysical parameters, are not influenced by the distance to Gallery 04.





**Table 4: Estimated geomechanical and geophysical parameters for the Opalinus Clay in the EZ-B niche normal and parallel to bedding based on the needle penetration index.**

| Parameter | Normal to bedding | Parallel to bedding | Reference |
|---|---|---|---|
| *NPI* | $98 \pm 29$ N mm$^{-1}$ $(n = 18)$ | $59 \pm 19$ N mm$^{-1}$ $(n = 29)$ | |
| Water content | 3.7 wt.-% | | |
| *UCS* | $29.7 \pm 12.5$ MPa | $18.2 \pm 8.0$ MPa | all UCS equations (average) |
| | $41.0 \pm 12.2$ MPa | $24.8 \pm 8.1$ MPa | Uchida et al. (2004) |
| | $19.6 \pm 5.8$ MPa | $11.9 \pm 3.9$ MPa | Aydan (2012) |
| | $28.3 \pm 7.7$ MPa | $17.7 \pm 5.4$ MPa | Ulusay and Erguler (2012) |
| *BTS* | $2.0 \pm 0.6$ MPa | $1.2 \pm 0.4$ MPa | Aydan et al. (2014) |
| *E* | $4.9 \pm 1.5$ GPa | $3.0 \pm 1.0$ GPa | Aydan et al. (2014) |
| *c* | $3.9 \pm 1.2$ MPa | $2.4 \pm 0.8$ MPa | Aydan et al. (2014) |
| $\varphi$ | $41.8 \pm 2.7$ ° | $36.8 \pm 2.8$ ° | Aydan et al. (2014) |
| $v_p$ | $3.3 \pm 0.4$ km s$^{-1}$ | $2.6 \pm 0.4$ km s$^{-1}$ | Aydan et al. (2014) |
| $v_s$ | $1.9 \pm 0.2$ km s$^{-1}$ | $1.5 \pm 0.2$ km s$^{-1}$ | Aydan et al. (2014) |





**Figure 6: Comparison of estimated geomechanical and geophysical parameters of the Opalinus Clay in the EZ-B niche of the Mont Terri URL (black symbols, this study) with literature data for the shaly facies, normal (a–g) and parallel (b–i) to bedding. In addition, a second NPT dataset of Blum et al. (2013) from various locations in the URL (shaly facies) is shown (gray symbols). Literature data (green and blue symbols) originate from other niches in the Mont Terri URL and are mainly adapted after Jaeggi and Bossart (2014) and references herein (the full list of source documents is provided in Appendix A).**





Based on the needle penetrometer data obtained in the EZ-B niche, the estimated uniaxial compressive strength ($UCS$) was
29.7 MPa for a water content of 3.7 wt.-% normal to bedding (Fig. 6a). For water contents below 4.5 wt.-%, no literature data
on $UCS$ is available for the shaly facies of the Opalinus Clay. Due to the fact that $UCS$ is increasing linearly with decreasing
water content (Wild et al., 2015), the estimated mean value can be considered reasonable and complements the literature
dataset in Fig. 6a. The additional NPT dataset of Blum et al. (2013) also confirms the negative linear correlation of $UCS$ and
water content. Parallel to bedding, the estimated $UCS$ of 18.2 MPa is consistent with literature data of the shaly facies for
similar water content (Fig. 6b). The two NPT datasets, acquired from the EZ-B niche as well as different locations in the Mont
Terri URL (Blum et al., 2013), exactly reproduce the trend that is evident from the available literature data. The estimated
$UCS$ values in Fig. 6a and b were obtained by using different empirical equations, leading to a comparatively high standard
deviation. The individual results for each empirical equation are listed in Table 4. As needle penetration testing is widely used
for estimating the $UCS$ of rocks (Ulusay et al., 2014), several functions are available which have been deduced from various
rock types (Table 2). For the Opalinus Clay, the applied combination of different established equations has proven to be
suitable.

The Brazilian tensile strength ($BTS$) of the Opalinus Clay normal to bedding could also be estimated by the needle penetration
tests. The mean value of 2 MPa that was obtained from the on-site measurements in the EZ-B niche is located approximately
in the center of the literature value range for a comparable water content (Fig. 6c). Apparently, the literature data for the shaly
facies is divided into two subgroups. Lower $BTS$ values normal to bedding are most likely related to pre-damaged sample
material, i.e. desiccation cracks, which preferentially form parallel to the bedding features in the drill core and therefore reduce
the tensile strength of the tested sample perpendicular to bedding. Parallel to bedding, the estimated $BTS$ deviates by about
2 MPa from the available literature data and is therefore underestimated (Fig. 6d).

Similar to $UCS$, the Young´s modulus $E$ is increasing monotonically with declining water content as evident from both
literature data and the NPT dataset of Blum et al. (2013). For water contents below 5.7 wt.-%, no literature data is available
for the shaly facies normal to bedding. As the derived Young´s modulus of 4.9 GPa normal to bedding for a water content of
3.7 wt.-% was significantly higher than the available literature values for natural water contents (maximum of 2.4 GPa), the
estimation was assumed to be applicable for the Opalinus Clay (Fig. 6e). With a mean value of 3 GPa, $E$ was largely
underestimated parallel to bedding compared to the available literature data ranging between 13 and 31 GPa for similar water
contents (Fig. 6f). This is most probably due to the fact that rather well preserved drill core specimens are compared to the
long exposed and highly damaged rock surface of a niche.

The elastic P-wave velocity ($v_p$) is slightly overestimated normal to bedding (Fig. 6g). Similar to Fig. 6c, two sub-datasets can
be identified within the literature data. Here, pre-damage of the core sample material or an insufficient coupling of the
ultrasonic source was most likely responsible for these reduced P-wave velocities, which are therefore not representative for
the intact rock body (Jaeggi and Bossart, 2014). According to Jaeggi and Bossart (2014), the core samples belonging to the
lower sub-dataset of literature values in Fig. 6g were partially penetrated by microcracks. The estimated $v_p$ data seems to



represent the upper subset of the literature data. Since the needle penetrometer only samples a small area on the rock surface, the NPT is able to reflect the actual in situ conditions. Parallel to bedding, $v_p$ is slightly underestimated but fairly represents the data range of the shaly facies (Fig. 6h). The observed trend of slight P-wave velocity increase with decreasing water content

of the Opalinus Clay normal and parallel to bedding is also well represented by the datasets obtained from needle penetration testing.

For the elastic S-wave velocity $v_s$ of the Opalinus Clay, a mean value of 1.9 km s$^{-1}$ was estimated normal to bedding. Due to strong absorption and attenuation of wave energy and insufficient signal retrieval during ultrasonic velocity measurement (Gräsle and Plischke, 2010; Schnier and Stührenberg, 2007), only few literature data is available. For the shaly facies of the

Opalinus Clay, Bock (2009) reports $v_s$ values of 1.51 km s$^{-1}$ for a natural water content of 6.4 wt.-%. Wileveau (2005) provides S-wave velocities of 1.45–1.58 km s$^{-1}$ for water contents varying between 2.4 and 2.9 wt.-%. Hence, normal to bedding $v_s$ is most likely overestimated by the NPT. Parallel to bedding, a value of 1.5 km s$^{-1}$ was determined. In comparison to existing literature data, $v_s$ is underestimated in this direction (Fig. 6i). As ultrasonic velocity is explicitly dependent on the internal structure of the rock, such as cementation, anisotropy and porosity structure (Jaeggi and Bossart, 2014; Schuster et al., 2017),

it can be assumed that the relationship of ultrasonic velocity and *NPI* is probably rather weak.

For the cohesion $c$ and the friction angle $\varphi$, reliability of the estimated values (Table 4) could not be assessed due to limited literature data, especially for samples with low water contents. In Bock (2009), mean values for cohesion (3.7 MPa and 5.4 MPa) and friction angle (22 ° and 23 °) of the Opalinus Clay for an average water content of 6.7 wt.-% are given normal and parallel to bedding, respectively. In contrast, Lisjak et al. (2015) used a cohesion of 24.8 MPa (normal) and 2.4 MPa (parallel)

as an input parameter for their calibrated finite-discrete element model. The implemented friction angle of 22 ° complies with Bock (2009), but no information on the assumed water content is included. Cohesion values of 3.9 MPa (normal) and 2.4 MPa (parallel) derived from needle penetrometer measurements in the EZ-B niche were of the same order of magnitude as the data given by Bock (2009). However, the estimated friction angles of 42 ° and 37 ° normal and parallel to bedding, respectively, were significantly higher in comparison to the literature data.

Generally, the applied empirical functions performed better normal to bedding. The poor estimation of parameters parallel to bedding (apart from *UCS*) may be linked to unperceived (possibly micro-scale) shrinking or desiccation cracks, which preferentially form parallel to the bedding features (e.g. Amann et al., 2017; Schnier and Stührenberg, 2007). Microcracks would facilitate needle penetration, thus reducing the measured *NPI* as well as the estimated parameter values. It should also be noted that the existing empirical relations for estimating physico-mechanical parameters do not specifically apply to

claystones or shales, but were derived based on compiled experimental data obtained for various types of soft rocks (Aydan et al., 2014). In addition to the factors mentioned above, sedimentary heterogeneity might also be responsible for geomechanical variability and the observed deviations between the *NPI*-based values and the considered literature data. However, heterogeneity of the shaly facies of the Opalinus Clay in Mont Terri is comparatively low (Jaeggi and Bossart, 2014). More likely, the observed deviations are due to variant surface constitution caused by the hugely varying exposure time of the rock



walls in the non-lined EZ-B niche in contrast to sampled specimens from drill cores. Although the measurements were conducted at the upper end of the specified application range of the needle penetrometer, physico-mechanical parameter estimation based on needle penetration testing can be recommended for indurated clays, especially for determining the anisotropic uniaxial compressive strength.

## 4 Conclusions

An excavation damaged zone (EDZ) in the Opalinus Clay of the Mont Terri Rock Laboratory, Switzerland, was characterized with regard to hydraulic and mechanical properties using three different methodological on-site approaches: (1) air permeameter, (2) microscope camera and (3) needle penetration test. About 15 years after excavation, artificially induced unloading joints (EDZ fractures), reactivated fault planes (tectonic fractures) and bedding-parallel desiccation cracks with a mean mechanical aperture of 233 µm and a mean hydraulic aperture of 84 µm were observed in the EZ-B niche, serving as

potential flow paths for advective transport in the indurated clay formation. This is not only limited to the area of the strongly pronounced EDZ around Gallery 04, where a dense network of interconnected fractures is encountered, but also applies to potentially reactivated tectonic discontinuities at greater distances, e.g. due to large-scale stress redistribution or injection of fluids. After an initial continuous aperture closure observed by long-term jointmeter data records in the non-lined niche, which can be attributed to seasonally controlled shrinkage and swelling cycles in combination with niche convergence, this process

seems to decelerate significantly after 15 years of monitoring. Locally, fractures are influenced by shotcrete application, leading to reduced hydraulic and mechanical apertures due to enhanced water availability and swelling of clay minerals in the immediate vicinity. Among the studied discontinuity types, the EDZ fractures showed the smallest hydraulic apertures. However, as 60 % of all measured values were within the range of 80–120 µm, a clear distinction was not possible.

From direct measurement with the portable transient-flow air permeameter, plausible hydraulic aperture data could be acquired

on-site, even if the entire range of fractures in the EZ-B niche could not be reproduced due to a limited measuring range. This means that the permeameter measurements tend to over represent smaller fractures. Here, indirect determination of hydraulic fracture apertures based on the automatic evaluation of high-resolution microscope camera images of fracture traces offers a practical alternative. Due to the smaller mean mechanical aperture of the artificially induced unloading fractures compared to the investigated tectonic fractures, conversion was most appropriate for the EDZ fractures. Tectonic fractures on average

exhibit a higher variance of measured distances along imaged fracture traces, which can be explained by a higher degree of mismatch between the fracture surfaces due to the reactivation of fault planes during excavation.

The needle penetration test proved to be a valuable tool, especially for accurate estimation of the anisotropic uniaxial compressive strength, as the needle penetration index satisfactorily reflects the in situ conditions of the intact rock mass. For the shaly facies of the Opalinus Clay in the EZ-B niche, a mean uniaxial compressive strength of 30 MPa (normal to bedding)

and 18 MPa (parallel to bedding) was determined. While parameter estimation based on needle penetration indices normal to bedding showed a high agreement with available literature data, physico-mechanical parameter values were mostly





underestimated in bedding-parallel direction. Due to damage of the exposed rock surface associated with the formation of microcracks parallel to stratification, needle penetration was facilitated in this case. Due to direct air contact and ventilation of the rock laboratory, the desaturation of the claystone in the near field of the niche led to a sharp decrease in water content to 3.7 wt.-%, which is directly linked to an increased uniaxial compressive strength, Young´s modulus and elastic P-wave velocity normal to bedding.

The applied on-site measurement methodology and evaluation approach provides a suitable instrument for the hydraulic and mechanical characterization of excavation damaged or disturbed zones in different geological environments, especially since drilling is not always feasible and the validity of estimated parameters is limited to the investigated location. With comparatively little effort, nondestructive analysis of time- and location-dependent variability of important parameters is permitted. Besides confirming the suitability of the methodological approach for flexibly determining hydraulic and mechanical properties, the study assesses the state of an EDZ in a non-lined niche after long-term exposure and therefore serves as an important guideline for diverse tunneling projects and future performance assessments of nuclear waste disposal sites in argillaceous rocks.

**Appendix A**

The literature data included in Fig. 6 (Sect. 3.2) were mainly taken from Jaeggi and Bossart (2014). This expert report offers a compilation of safety related rock parameters determined for the Opalinus Clay in the Mont Terri URL. All source documents for experimentally derived data on geomechanical and geophysical properties that were utilized in this study are listed in Table A1. Technical Notes (TN) and Technical Reports (TR) of the Mont Terri Project are accessible via https://www.mont-terri.ch/en/documentation/technical-reports.html.



**Table A1: Source documents for experimental data on uniaxial compressive strength ($UCS$), Brazilian tensile strength ($BTS$), Young´s modulus ($E$) and elastic P-wave ($v_P$) and S-wave velocity ($v_S$) of the Opalinus Clay (shaly facies) in Mont Terri.**

| Source document | $UCS$ | $BTS$ | $E$ | $v_P$ | $v_S$ |
|---|:---:|:---:|:---:|:---:|:---:|
| Amann et al. (2011) | ● | | ● | ● | |
| Amann et al. (2012) | | | | ● | |
| Gräsle and Plischke (2010), TR | ● | ● | ● | | ● |
| Gräsle and Plischke (2011), TN | | | | ● | |
| Jahns (2010), TN | ● | ● | | | |
| Rummel and Weber (2004), TN | ● | | | ● | |
| Rummel and Weber (2007), TN | ● | | | ● | |
| Schnier and Stührenberg (2007), TR | ● | | | | ● |
| Soe et al. (2009) | | | | | ● |
| Wild (2010) [a] | | ● | | | |
| Wild et al. (2015) | ● | ● | ● | ● | |
| Wileveau (2005), TR | | | | ● | ● |
| Wymann (2013) [a] | ● | | ● | ● | |
| Zimmer (2012) [a] | | ● | | ● | |

[a] unpublished final theses, ETH Zurich, Switzerland

**Code and data availability**

All field data related to this manuscript and the MATLAB code for the evaluation of microscopic fracture trace images is available at https://doi.org/10.6084/m9.figshare.12581144.v2 (Hale et al., 2020a).

**Author contribution**

SH, XR and PB carried out the measurements in the Mont Terri URL, DJ handled the organization and implementation of the field work. Formal analysis was done by SH and XR. PB supervised SH and XR and was responsible for funding acquisition.
SH wrote the initial draft and all authors (SH, XR, DJ and PB) discussed and interpreted the results and substantially contributed to editing and reviewing the manuscript.



**Competing interests**

The authors declare that they have no conflict of interest.

**Acknowledgements**

We thank swisstopo, the Federal Office of Topography in Switzerland, for enabling and supporting our field work at the Mont
Terri Rock Laboratory and for providing valuable documentation. This work was financially supported by the German Federal
Ministry of Education and Research (BMBF) "Geological Research for Sustainability (GEO:N)" program [grant no.
03G0871D] within the framework "Research for Sustainable Development (FONA3)".

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
