# Peer review of "Mechanical and hydraulic properties of the excavation damaged zone (EDZ) in the Opalinus Clay of the Mont Terri Rock Laboratory, Switzerland"

_Solid Earth, 2021_

## Referee Comment (RC2)

*Mechanical and hydraulic properties of the excavation damaged zone (EDZ) in the Opalinus Clay of the Mont Terri Rock Laboratory, Switzerland*

*Sina Hale, Xavier Ries, David Jaeggi, Philipp Blum*

**General comments**

**Research question**

The current research paper addresses the question of how to easily, cost-effectively and non-destructively investigate the (time-dependent) evolution of a discrete fracture network (DFN) in an excavation damaged zone (EDZ) of an unlined lateral adit to draw conclusions for storage safety of e.g. long-term nuclear storage waste disposal. The usefulness of the proposed set of partially in-situ methods is shown by a comparison of the authors own results based on an acquired dataset in a small, unlined niche of the Mont Terri URL in Switzerland with the vast data repository from numerous scientific investigations conducted in the same overall location since 1996 (=25 years).

The authors present a four-fold dataset (measured in April 2019) consisting of:

- (43) direct measurements of hydraulic fracture aperture $a_h$ using an air permeameter on rough fracture surfaces;
- (36) estimations of both mechanical fracture aperture $a_m$ coupled with fracture surface roughness $\delta a_m$ by the use of image analysis on a set of microscopic images on rough fracture surfaces (methodology and an automation thereof recently published by the main author);
- (47) direct measurements of needle penetration indices (NPI) perpendicular and parallel to bedding on rough fracture surfaces;
- (2) indirect measurements of the water content of an opalinus clay sample.

The set of estimated mechanical aperture $a_m$ values and the fracture surface roughness $\delta a_m$ values derived from image analysis are then related to equivalent hydraulic fracture apertures $a_{h,eq}$ via a cubic law additionally requiring single fracture transmissivities $T_f$ measured at Mont Terri URL in other studies for plausibility purposes. The set of of NPI measurements are subsequently split into two categories (parallel and perpendicular to bedding) and a range of geomechanical and geophysical parameters calculated on a range of empirical relationships readily available in literature. Together with the determined water content of a representative sample the calculated parameters are then related to the ones available by other studies from Mont Terri data repository.

Through a classification of the investigated fractures as artificially induced unloading fractures (called "EDZ fractures"), reactivated fault planes/splays (called "tectonic fractures") and bedding-parallel desiccation/unloading cracks the directly measured and derived data is then set into context of the temporal development of the EDZ in the investigated niche and to the whole Mont Terri URL project.

*Contribution of work*

Presentation quality

The presentation quality of the paper is considered Excellent (1) as it is very well structured, nicely illustrated (plots and figures) and clear by not mixing its own observations with already existing and new interpretations. The abstract is of very good quality. The use of the English language is concise, does not suffer from any noticeable flaws and therefore overall makes a joyful read.

Scientific quality

The scientific quality of the current paper is assessed as Good (2) as it uses well-established approaches (i.e. air permeameter) and available data from the site as well as the vast knowledge of the Mont Terri URL site to accurately assess the performance and limitations of the authors own microimage-methodology and its applicability to similar investigations.

Overall scientific significance

The question to find an easy-to-apply methodology to characterize the time-dependent evolution of a DFN in an EDZ is interesting as well as important. The investigation of the bulk material (i.e. different facies of opalinus clay) and derived bulk chemical and physical properties as well as effective medium approximations only show the side of the story where material behavior is controlled by the bulk medium only. Although the bulk material properties may be very favorable, very local discontinuities created during excavation ("EDZ fractures") and first and foremost local to regional (re-activated) tectonic discontinuities and the transport properties along them very likely play – apart from the above mentioned bulk material considerations - a very important role in the assessment of storage safety in the host rock.

Scientific significance of the paper

The main contribution of this research lies in the comparison of the acquired dataset of hydraulic fracture apertures $a_h$ determined by an commercial of-the-shelf air permeameter with the mechanical fracture aperture $a_m$ coupled with fracture surface roughness $\delta a_m$ determined by the image analysis approach of the first author (i.e. Hale et al., 2020b). As the conversion of geometrically measured parameters ($a_m$, $\delta a_m$) to hydraulic fracture apertures $a_h$ relies on empirical relationships, plausibility of the converted values – as mentioned by the authors – rely on fracture transmissivities $T_f$ measured by other methods (e.g. borehole geophysics) in the same overall location. Therefore, the current study as a use case for the first authors own microimage-methodology accurately depicts its main advantages (i.e. measurement range, spatial resolution) but also its own main limitation mainly due to reliance on empirical relationships as well as due to the related error propagation of large standard deviations due to variable fracture surface roughness.

If looked at from the angle of a use case of an already published methodology paper, the current research paper deserves its merit – coupled with independently acquired direct measurements of $a_h$ or via $T_f$– as an incremental advance in the field of non-destructive, cost-effective, in-situ determination of hydraulic fracture parameters. The microimage-method is – as indirectly pointed out by the authors – however not a standalone method for the determination of hydraulic fracture parameters (=geometric method) that can be used independently of direct physical methods (=kinematic methods).

If the research paper is however considered as its wrapping suggests (i.e. the advancement of the understanding of EDZ in opalinus clay for long-term nuclear storage), it falls short of its aim of significantly extending the knowledge by its new dataset provided compared to the one

that is already available from the investigated site and conclusion already drawn in previous studies (i.e. knowledge of seasonal cycles, inhibition of long-term convergence, significance of discontinuity types, dependency of strength on water content, problems with low humidity environment, shotcrete application to counter dehydration etc.). The determination of NPI as an extension of the papers dataset does – in the eye of the reviewer – by itself not extend the scientific knowledge of the strength anisotropy of (altered) layered geologic media (e.g. opalinus clay) as similar methods (e.g. Schmidt hammer) have readily been used by engineering geologists and geotechnical engineers for decades.

The scientific significance of the paper is therefore assessed as Good (2) if considered as a use case for Hale et al. (2020b) but overall as Fair (3) in its current form and wrapping and in the scope of the journal Solid Earth (SE).

Summary

Although the manuscript is very well built and structured as well as very nicely and clearly presented, it unfortunately falls short by not producing an overall dataset that adds to the already vast data repository of Mont Terri URL and does by itself not offer new interpretations and conclusions that significantly add to the understanding the temporal evolution of EDZ in opalinus clay.

At the bottom line, it is therefore questionable in the eye of the reviewer if the current manuscript represents a substantial contribution to scientific progress in the scope of Solid Earth but would rather be suited to a more specific journal.

**Specific comments - Questions/Issues**

Positive

- Setting the scene for the significance of the overall research question in the introduction by comparing bulk rock considerations vs. different behavior of EDZs;
- Relating the later measured hydraulic fracture apertures to the main hydraulic fracture parameters;
- Establishing the chosen methods "permeameter" + "microimaging" as easy to carry out in-situ alternatives to lab methods of borehole geophysics;
- Good characterization of the study site (2.1) with detailed references;
- Suitability of air permeameter measurements for the current study with its compatibility for advective flow;
- Comprehensive and well-written description of image analysis method (Section 2.3);
- Table 1 und Table 2 for the relevant conversion of measured parameters to calculated parameters later on;
- Structural maps of the niche with data plot and KDE – very nicely illustrated!
- Figure 5: very nicely illustrated key plot of the study;
- Line 370: Clear and honest statement that direct measurement of $a_h$ is preferable (variety of empiric relationships; error propagation etc) but microimaging offers additional geometric considerations that the hydraulic aperture does not;
- Conclusion: Fracture traces indeed offer a complementary (but not alternative) method to indirectly look at hydraulic fracture apertures;
- The "methodological trinity" indeed offers an interesting low-cost but somewhat limited approach to look at the temporal evolution of fractures in an EDZ.

Shortcomings

- Although the methodological "trinity" (i.e. air permeameter, microimaging, NPI-testing) is praised for its suitability of time-dependent monitoring of EDZ, it is however in the current study only applied at one point in time and then somehow related to the initial state after excavation. Therefore no temporal evolution is shown but only inferred by comparison to other studies;
- Schmidt hammers have been used by geologists in the industry for much longer than indicated by the authors (Line 111; oldest reference from 2007);
- Line 232: Were only nine (9) EDZ fractures measured? If yes, it is hard to make a significant statement based on a statistic comprising only nine values;
- Line 366: Related to Line 232: if only nine (9) EDZ fractures were measured, what significance does the statement have?
- Line 388ff: The data does not exhibit new findings but rather underlines existing knowledge (i.e. strength dependency on water content and fabric);
- Figure 6: although very nicely illustrated again, the addition of only one (1) dot to the compiled plots does not really underline the significance within the range of the all the data available;
- Section 3.2 does not offer a directly measured new dataset but only a calculated set of data points that does not offer room for a new interpretation. All the mentioned findings have already been reported in the (always properly referenced) literature;
- Conclusion: There is not enough data to make comparative statement of EDZ fractures (n=9!) vs. Tectonic Fractures (n=31) that would underline a statistically significant trend.

**Technical Corrections**

- Line 34: Is EdZ used as a synonym for EDZ?
- Line 46: Please introduce the expression THMC for the general audience;
- Line 139: EDZ fractures – is that distance measured along the niche or radial to the niche?
- Line 178: Do you mean "arithmetic mean" or "geometric mean"?

---

## Author Comment (AC1)

**Responses to comments by reviewer #1**

**Zeynal Abiddin Erguler (Referee)**

Ref: se-2021-4-RC1

Dear Authors,

Please be informed that I read carefully your manuscript. I would like to say that the content and outputs of your research are very important for scientists investigating fracture closure and self-sealing of mudrocks. I have only below given minor comments on your manuscripts:

**Response:** We sincerely thank Zeynal Abiddin Erguler for his constructive comments and valuable suggestions, which greatly helped to improve the manuscript's quality. In the following, we provide a point-by-point response to the comments, where comments are in black, and our responses are in blue. In addition, any changes regarding ''author responses to reviewer #1'' applied in the revised manuscript are also marked.

(1) Lines 16 and 403: The needle penetrometer cannot be used for geophysical characterization of rock materials. Only mechanical properties can be predicted by this index test. I suggest authors do not trust too much on predicted geophysical values and so the related empirical approaches of previous studies. Actually, you do not need these predicted parameters for this manuscript. Please apply proper modification.

**Response:** We agree. However, Aydan et al. (2014) demonstrated that ultrasonic P- and S-wave velocities can be roughly estimated by the NPI ($R^2 = 0.76$ and $R^2 = 0.76$, respectively), which was derived empirically from various tested lithologies.

Hence, we also applied the published empirical relationships in our study, compared and discussed the results in context to other studies in the literature. Finally, we were able to derive the following conclusion (page 23, line 464 in the marked manuscript):

*"[...] it can be assumed that the relationship of ultrasonic velocity and NPI is probably rather weak."*

This statement actually supports the reviewer´s comment. Hence, we would like to keep this information in our manuscript as we also intend to demonstrate the limitations of needle penetrometer testing (NPT), which is at least valid for the studied claystone.

(2) The introduction part is very long and so it is very hard to see the main motivations of this study. This part should be shortened. Since there are many previous studies on physical and hydro-mechanical properties of Opalinus Clay, the contribution of this study should be more strong to convince readers to read the entire paper.

**Response:** We also agree. The introduction was therefore shortened in the marked manuscript. In order to emphasize the novelty and contribution as well as the scientific significance of the study more clearly, the last section of the introduction was adapted as follows:

*"Hence, the objective of this study is to investigate the hydro-mechanical properties of the EDZ in the Opalinus Clay of the Mont Terri URL using on-site measurements on the exposed rock surface. In this study, a nondestructive and holistic determination of hydraulic and mechanical parameters of the fractured rock mass around a small tunnel is conducted by applying a combined approach using a transient-flow air permeameter, a microscope camera and a needle penetration test. Beside the bulk rock properties of the claystone, mechanical and hydraulic apertures of different fracture types of*

*the EDZ are quantified and discussed in this study, since these discontinuities can significantly control the overall material behavior. Furthermore, alterations within the EDZ of a non-lined niche due to several years of direct air exposure of the rock surface are investigated and discussed."*

(3) Some of relationships given in Figure 6 (Fig 6c, d, f, g, h, and i) are not statistically significant. I recommend removing them.

**Response:** We partially agree. For some of the parameters shown in Figure 6, the deviation of the NPI derived values from the existing literature data sets is very large. Particularly for the determination of the Brazilian tensile strength and the Young's Modulus parallel to bedding, the needle penetrometer test results in a significant underestimation of the respective values, which was explained by the presence of bedding-parallel microcracks.

However, this information is essential for the application of this method to the investigated host rock (and possibly also for other anisotropic claystone formations in future investigations). Thus, the mentioned relationships between the NPT data and the literature data in Fig. 6 (subfigures c, d, f, g, h and i) should also be retained from our point of view. In the last part of Section 3.2, however, it is clearly stated that for the Opalinus Clay the method is mainly suitable for determining the uniaxial compressive strength and that bedding-parallel measurements can lead to a poor estimation (page 23, line 474 in the marked manuscript).

(4) The manuscript looks very long. It would be a little difficult to read without getting bored for those who do not work on this subject. Please shorten your manuscript by removing unnecessary evaluations, discussion and outputs of previous studies.

**Response:** We agree. Apart from the introduction according to comment (2) above, also some discussion parts of Section 3 were shortened (page 14, lines 258-273 and page 15, lines 294-301 in the marked manuscript).

Except for the discussion on predicted P-wave and S-wave velocities, the content of this manuscript looks interesting. So I recommend minor revision for this submission.

Regards

Zeynal

---

## Author Comment (AC2)

**Responses to comments by reviewer #2**

**Anonymus Referee #2 (Referee)**

Ref: se-2012-4-RC2

**General comments**

Research question

The current research paper addresses the question of how to easily, cost-effectively and non-destructively investigate the (time-dependent) evolution of a discrete fracture network (DFN) in an excavation damaged zone (EDZ) of an unlined lateral adit to draw conclusions for storage safety of e.g. long-term nuclear storage waste disposal. The usefulness of the proposed set of partially in-situ methods is shown by a comparison of the authors own results based on an acquired dataset in a small, unlined niche of the Mont Terri URL in Switzerland with the vast data repository from numerous scientific investigations conducted in the same overall location since 1996 (=25 years). The authors present a four-fold dataset (measured in April 2019) consisting of:

- (43) direct measurements of hydraulic fracture aperture $a_h$ using an air permeameter on rough fracture surfaces;
- (36) estimations of both mechanical fracture aperture $a_m$ coupled with fracture surface roughness $\delta a_m$ by the use of image analysis on a set of microscopic images on rough fracture surfaces (methodology and an automation thereof recently published by the main author);
- (47) direct measurements of needle penetration indices (NPI) perpendicular and parallel to bedding on rough fracture surfaces;
- (2) indirect measurements of the water content of an opalinus clay sample.

The set of estimated mechanical aperture $a_m$ values and the fracture surface roughness $\delta a_m$ values derived from image analysis are then related to equivalent hydraulic fracture apertures $a_{h,eq}$ via a cubic law additionally requiring single fracture transmissivities $T_f$ measured at Mont Terri URL in other studies for plausibility purposes. The set of of NPI measurements aresubsequently split into two categories (parallel and perpendicular to bedding) and a range of geomechanical and geophysical parameters calculated on a range of empirical relationships readily available in literature. Together with the determined water content of a representative sample the calculated parameters are then related to the ones available by other studies fromMont Terri data repository.

Through a classification of the investigated fractures as artificially induced unloading fractures (called "EDZ fractures"), reactivated fault planes/splays (called "tectonic fractures") and bedding-parallel desiccation/unloading cracks the directly measured and derived data is then set into context of the temporal development of the EDZ in the investigated niche and to the whole Mont Terri URL project.

**Response**: We sincerely thank the anonymous Referee #2 for his or her detailed and constructive comments, which helped to substantially improve the manuscript. In the following, we provide a point-by-point response to the comments, where comments are in black, and our responses are in blue. In addition, any changes regarding ''author responses to reviewer #2'' are highlighted in the marked revised manuscript.

**Contribution of work**

(1) Presentation quality: The presentation quality of the paper is considered Excellent (1) as it is very well structured, nicely illustrated (plots and figures) and clear by not mixing its own observations with already existing and new interpretations. The abstract is of very good quality. The use of the English language is concise, does not suffer from any noticeable flaws and therefore overall makes a joyful read.

(2) Scientific quality: The scientific quality of the current paper is assessed as Good (2) as it uses well-established approaches (i.e. air permeameter) and available data from the site as well as the vast knowledge of the Mont Terri URL site to accurately assess the performance and limitations of the authors own microimage-methodology and its applicability to similar investigations.

(3) Overall scientific significance: The question to find an easy-to-apply methodology to characterize the time-dependent evolution of a DFN in an EDZ is interesting as well as important. The investigation of the bulk material (i.e. different facies of opalinus clay) and derived bulk chemical and physical properties as well as effective medium approximations only show the side of the story where material behavior is controlled by the bulk medium only. Although the bulk material properties may be very favorable, very local discontinuities created during excavation ("EDZ fractures") and first and foremost local to regional (re-activated) tectonic discontinuities and the transport properties along them very likely play – apart from the above mentioned bulk material considerations - a very important role in the assessment of storage safety in the host rock.

**Response:** We appreciate the positive assessment of the quality of the submitted manuscript and of the overall scientific significance of the research question.

(4) Scientific significance of the paper: The main contribution of this research lies in the comparison of the acquired dataset of hydraulic fracture apertures $a_h$ determined by an commercial of-the-shelf air permeameter with the mechanical fracture aperture $a_m$ coupled with fracture surface roughness $\delta a_m$ determined by the image analysis approach of the first author (i.e. Hale et al., 2020b). As the conversion of geometrically measured parameters ($a_m$,$\delta a_m$) to hydraulic fracture apertures $a_h$ relies on empirical relationships, plausibility of the converted values – as mentioned by the authors – rely on fracture transmissivities $T_f$ measured by other methods (e.g. borehole geophysics) in the same overall location. Therefore, the current study as a use case for the first authors own microimage-methodology accurately depicts its main advantages (i.e. measurement range, spatial resolution) but also its own main limitation mainly due to reliance on empirical relationships as well as due to the related error propagation of large standard deviations due to variable fracture surface roughness. If looked at from the angle of a use case of an already published methodology paper, the current research paper deserves its merit – coupled with independently acquired direct measurements of $a_h$ or via $T_f$ – as an incremental advance in the field of non-destructive, cost-effective, in-situ determination of hydraulic fracture parameters. The microimage-method is – as indirectly pointed out by the authors – however not a standalone method for the determination of hydraulic fracture parameters (=geometric method) that can be used independently of direct physical methods (=kinematic methods).

If the research paper is however considered as its wrapping suggests (i.e. the advancement of the understanding of EDZ in opalinus clay for long-term nuclear storage), it falls short of its aim of significantly extending the knowledge by its new dataset provided compared to the one that is already available from the investigated site and conclusion already drawn in previous studies (i.e. knowledge of seasonal cycles, inhibition of long-term convergence, significance of discontinuity types, dependency of strength on water content, problems with low humidity environment, shotcrete application to counter dehydration etc.). The determination of NPI as an

extension of the papers dataset does – in the eye of the reviewer – by itself not extend the scientific knowledge of the strength anisotropy of (altered) layered geologic media (e.g.opalinus clay) as similar methods (e.g. Schmidt hammer) have readily been used by engineering geologists and geotechnical engineers for decades. The scientific significance of the paper is therefore assessed as Good (2) if considered as a use case for Hale et al. (2020b) but overall as Fair (3) in its current form and wrapping and in the scope of the journal Solid Earth (SE).

**Response:** We partially disagree. Since comment (5) also partly questions the scientific significance of the manuscript, we would like to refer to our response to comment (5), where we explained the contribution and relevance of the presented study in more detail.

(5) Summary: Although the manuscript is very well built and structured as well as very nicely and clearly presented, it unfortunately falls short by not producing an overall dataset that adds to the already vast data repository of Mont Terri URL and does by itself not offer new interpretations and conclusions that significantly add to the understanding the temporal evolution of EDZ in opalinus clay. At the bottom line, it is therefore questionable in the eye of the reviewer if the current manuscript represents a substantial contribution to scientific progress in the scope of Solid Earth but would rather be suited to a more specific journal.

**Response**: First and foremost, the applied combination of three different non-destructive measurement methods is absolutely novel.

The presented methodical-analytical approach was successfully used for a holistic hydro-mechanical characterization of an EDZ in a potential host rock formation for nuclear waste disposal. Thus, the presented approach is certainly of great importance for future investigations, since required parameters of the bulk rock and the fracture network can be determined accurately using only simple measuring methods.

Due to the fact that the new combined approach was compared with pre-existing data, the measured values yielded conclusions that are already known from previous studies (significant anisotropy of the Opalinus Clay, dependence of strength and stiffness on the water content, influence of the saturation state on fracture permeability or aperture in the EDZ). In addition, however, other important results and new interpretations emerged from this study that are relevant both for the characterization of host rocks for nuclear waste disposal and for the characterization of EDZs in tunnels:

- The Opalinus Clay in the vicinity of a tunnel (on site) actually behaves as assumed on the basis of the extensive laboratory tests on smaller core samples.
- Determination of mechanical and hydraulic fracture apertures with an advanced form of the microimaging method developed by the first author. In contrast to the publication of Hale et al. (2020) mentioned by the reviewer, the method was automated to enable characterization of a higher number of fractures within short time. In addition, the submitted manuscript shows that microimaging of fracture traces also enables further geometric considerations on formation mechanisms.
- Regardless of the lithology (e.g. sandstone (Hale et al., 2020), claystone (this study)), it is shown that the equation from Kling et al. (2017) is most suitable for the estimation of the hydraulic aperture ($a_h$) based on fracture trace imaging and can therefore be highly recommended for future use. It is also shown that the median is the most suitable parameter to provide a representative measure of the mechanical aperture of a fracture network.
- Air permeameters are primarily used for determining the matrix permeability of rocks. We extend the application range of the measurement method to a discrete fracture network (DFN) of an EDZ within the scope of nuclear waste disposal.

- When applying the needle penetrometer test (NPT) to determine geomechanical properties in bedding-parallel direction, the respective parameters can be significantly underestimated. This finding was explained by the presence of bedding-parallel microcracks and may therefore also apply to other anisotropic claystone formations.
- The study yields therefore several new data sets:
  - Various geomechanical parameters of the Opalinus Clay (shaly facies) for various water contents obtained by a measurement method not previously used for this lithology. It should also be mentioned that these data reflect a 'naturally' adjusted saturation state without artificial drying or saturation (several years of exposure).
  - Mechanical and hydraulic fracture apertures ($a_h$, $a_m$) of different EDZ fracture types in the Opalinus Clay. To our knowledge, there is no such data set available yet.
  - Extensive compilation of existing physico-mechanical literature data. To our knowledge, such a compilation in its presented form has not been published yet. This data compilation could be used for modeling that also includes, for example, sensitivity analyses, as the range of values for different important parameters can be identified and quantified.

The submission of the manuscript to the journal Solid Earth with its interdisciplinary scope is supported by the fact that the presented methods and results can also be applied and transferred to different research topics and questions addressed by the journal.

In order to emphasize the contribution of the study as well as the scientific significance of the manuscript more clearly, the last section of the introduction was adapted (page 4, lines 117-125 in the marked manuscript, see Response to comment (2) of Reviewer #1).

**Specific comments – Questions/Issues**

(6) Positive:

- Setting the scene for the significance of the overall research question in the introduction by comparing bulk rock considerations vs. different behavior of EDZs;
- Relating the later measured hydraulic fracture apertures to the main hydraulic fracture parameters;
- Establishing the chosen methods "permeameter" + "microimaging" as easy to carry out in-situ alternatives to lab methods of borehole geophysics;
- Good characterization of the study site (2.1) with detailed references;
- Suitability of air permeameter measurements for the current study with its compatibility for advective flow;
- Comprehensive and well-written description of image analysis method (Section 2.3);
- Table 1 und Table 2 for the relevant conversion of measured parameters to calculated parameters later on;
- Structural maps of the niche with data plot and KDE – very nicely illustrated!
- Figure 5: very nicely illustrated key plot of the study;
- Line 370: Clear and honest statement that direct measurement of ah is preferable (variety of empiric relationships; error propagation etc) but microimaging offers additional geometric considerations that the hydraulic aperture does not;
- Conclusion: Fracture traces indeed offer a complementary (but not alternative) method to indirectly look at hydraulic fracture apertures;
- The "methodological trinity" indeed offers an interesting low-cost but somewhat limited approach to look at the temporal evolution of fractures in an EDZ.

**Response:** We are pleased that so many points stood out positively while reviewing our manuscript.

(7) Shortcomings:

- Although the methodological "trinity" (i.e. air permeameter, microimaging, NPI-testing) is praised for its suitability of time-dependent monitoring of EDZ, it is however in the current study only applied at one point in time and then somehow related to the initial state after excavation. Therefore no temporal evolution is shown but only inferred by comparison to other studies;

**Response:** The aim of the study was to characterize an EDZ in a simple but effective way with regard to its hydro-mechanical properties. In the submitted manuscript, only in the Abstract (page 1, line 23) and in the Conclusion (page 25, line 520) it was pointed out that this combination of methods is in principle suitable for a time-dependent monitoring of an EDZ. Since the hydraulic properties of the discrete fracture network and the mechanical properties of the bulk rock can be determined non-destructively and, in particular, without introducing artificial changes to the system, the same section of an EDZ can be tested repeatedly over a longer period of time.

- Schmidt hammers have been used by geologists in the industry for much longer than indicated by the authors (Line 111; oldest reference from 2007);

**Response:** We agree that the Schmidt hammer and also the needle penetrometer test (NPT) has been used for a longer period of time than the cited references imply. We have therefore added the following references (page 4, lines 111-112 in the marked manuscript):

*Hucka, V.: A rapid method of determining the strength of rocks in situ, Int. J. Rock Mech. Min. Sci. & Geomech. Abstr., 2, 127–134, https://doi.org/10.1016/0148-9062(65)90009-4, 1965.*

*Okada, S., Izumiya, Y., Iizuka, Y., and Horiuchi, S.: The estimation of soft rock strength around a tunnel by needle penetration test, J Jpn Soc Soil Mech Found Eng, 33, 35–38, 1985.*

- Line 232: Were only nine (9) EDZ fractures measured? If yes, it is hard to make a significant statement based on a statistic comprising only nine values;

**Response:** The information given on page 11, line 232 in the submitted manuscript relates to all measurable fractures in the niche, which are affected by the EDZ of the EZ-B niche (reactivated and newly formed discontinuities). The information given in line 234 explicitly refers to the newly formed "EDZ fractures" (artificially induced unloading joints). Due to the subsequent application of shotcrete in the entrance area of the niche, less EDZ fractures were available for measurement compared to the tectonic fractures.

For clarity, we specified the respective number of measurements for each fracture type (lines 237-243 in the marked manuscript).

- Line 366: Related to Line 232: if only nine (9) EDZ fractures were measured, what significance does the statement have?

**Response:** The observation was made using the available data and was therefore addressed in this context, although the number of measurements of the two fracture types was different. However, also with regard to the last comment, we pointed out the limited number of measured values here (page 18, lines 372-373 in the marked manuscript):

"*While EDZ fractures seem to better correspond to the general model concept of an "ideal plane-parallel fracture", tectonic fractures are most probably characterized by a different $a_m$–$a_h$–relation.*

*While implied by the presented measurement data, this issue should still be examined based on a larger data set."*

- Line 388ff: The data does not exhibit new findings but rather underlines existing knowledge (i.e. strength dependency on water content and fabric);
- Figure 6: although very nicely illustrated again, the addition of only one (1) dot to the compiled plots does not really underline the significance within the range of the all the data available;
- Section 3.2 does not offer a directly measured new dataset but only a calculated set of data points that does not offer room for a new interpretation. All the mentioned findings have already been reported in the (always properly referenced) literature;

**Response:** We partially disagree with the last three comments. In Section 3.2 of the submitted manuscript we show that the needle penetrometer is suitable for the Opalinus Clay and provides reliable results, which was enabled by a comparison of the calculated values based on the needle penetration index (NPI) with compiled literature data sets. The agreement of the new data set with the existing data and knowledge clearly confirms that the needle penetrometer estimation expresses the anisotropy as well as the strength properties of the Opalinus Clay, although only a mm-sized area of the rock surface is tested by the device.

Section 3.2 actually includes a new data set based on a measurement and estimation procedure that differs from all existing literature data. One of the authors was also involved in the creation of the NPI dataset from 2013, but these results have not been compared to other data yet and are therefore revisited here. Overall, this results in a needle penetrometer data set for the shaly facies of the Opalinus Clay for nine different saturation states with water contents varying between 0 wt.% and 6.5 wt.%.

In addition, for some of the mechanical parameters considered, the compiled literature data set was also extended. In particular, for the uniaxial compressive strength (UCS) and the Young´s modulus (E), no measured values obtained by conventional methods were available for low water contents (< 5 wt.%) normal to bedding.

- Conclusion: There is not enough data to make comparative statement of EDZ fractures (n=9!) vs. Tectonic Fractures (n=31) that would underline a statistically significant trend.

**Response:** We agree, but we are convinced that the data and the conclusion is important for the reader and therefore we would like to keep this conclusion. In addition, the presented comparative examination including both geometric and hydraulic characteristics of different fracture types can provide a starting point for future studies.

**Technical Corrections**

- Line 34: Is EdZ used as a synonym for EDZ?

**Response:** EdZ is not used as a synonym for EDZ. The abbreviation EDZ denotes the excavation *damaged* zone, where significant changes in flow and transport properties are observed. EdZ designates the excavation *disturbed* zone, where the hydraulic properties of the rock are only scarcely affected. These abbreviations were already defined in the submitted manuscript (page 1, lines 27-28 and page 2, lines 30-35).

- Line 46: Please introduce the expression THMC for the general audience;

**Response:** The term THMC (thermal, hydrological, mechanical and chemical) was introduced accordingly (page 2, line 47 in the marked manuscript).

– Line 139: EDZ fractures – is that distance measured along the niche or radial to the niche?

**Response:** The specified distance refers to the depth along the niche, where EDZ fractures originating from the excavation of the adjacent Gallery 04 are present. For clarification, the respective sentence was modified (page 5, line 142 in the marked manuscript):

*"Excavation-induced unloading joints (EDZ fractures) that are related to the construction of Gallery 04 are present within the first 1.3 m along the EZ-B niche (Nussbaum et al., 2005)."*

– Line 178: Do you mean "arithmetic mean" or "geometric mean"?

**Response:** We were indeed referring to the arithmetic mean here, as already stated in the submitted manuscript.

---

## Author Comment (AC3)

**Responses to Topical editor recommendations**

**Virginia Toy (Topical editor)**

Ref: se-2012-4-EC1

I have read over the referee comments, and your responses, and also checked the changes you made to the manuscript. I mostly think you've done a good job of addressing their concerns, and I find the manuscript to be scientifically sound and worthy of publication.

**Response**: We thank the Topical editor Virginia Toy for her recommendations, which are reasonable and helpful. In the following, we provide a point-by-point response to each comment, where comments are in black, and our responses are in red. In addition, the changes regarding ''author responses to Topical editor recommendations'' are highlighted in red in the marked revised manuscript, while the previous changes related to the reviewers' comments are highlighted in blue.

However, there are a couple of places where I recommend you do a little bit more work to adequately address the reviewer's comments and further improve the manuscript, as follows:

I would re-write your revised intro sentences as follows:

"Hence, the objective of this study was to investigate the hydro-mechanical properties of the EDZ in the Opalinus Clay of the Mont Terri URL from in situ measurements on the exposed rock surface. We carried out a nondestructive and holistic determination of hydraulic and mechanical parameters of the fractured rock mass around a small tunnel, by combining transient-flow air permeametry, photomicroscopy, and needle penetration tests. We characterised bulk rock properties of the claystone, and quantified mechanical and hydraulic apertures of different fracture types of the EDZ, since these discontinuities can significantly control the overall material behavior. We have also explored alteration of a non-lined niche that was directly exposed to air for several years …. *and then say WHY and HOW you did this.*

**Response**: We agree. Thank you for the suggested changes to the introductory sentences of the manuscript. We have mostly adopted the wording given above and have expanded it accordingly:

*"Hence, the objective of this study was to investigate the hydro-mechanical properties of the EDZ in the Opalinus Clay of the Mont Terri URL from in situ measurements on the exposed rock surface. We carried out a nondestructive and holistic determination of hydraulic and mechanical parameters of the fractured rock mass around a small tunnel niche, by combining transient-flow air permeametry, photomicroscopy and needle penetration tests. We characterized bulk rock properties of the claystone, and quantified mechanical and hydraulic apertures of different fracture types of the EDZ, since these discontinuities can significantly control the overall material and flow behavior. We have also explored the alteration of the non-lined niche that was directly exposed to air for several years. By using the water content of the claystone, we compared the determined physico-mechanical parameters with data from other studies to assess the effect of desaturation directly on-site at the tunnel wall."*

Reviewer 1 criticised inclusion of discussion of the P and S wave velocity. You did not respond to their comment. Please provide some sort of response.

**Response**: We already addressed this point in our response to comment (1) of reviewer #1. Here we have outlined that the empirical relations we used to estimate the P- and S-wave velocities are existing published equations, which led to satisfactory results in the study of Aydan et al. (2014) and can in principle also be used for the investigated claystone.

In agreement with reviewer #1, however, we also found that the estimation procedure based on the needle penetrometer tests is less suitable for determining P- and S-wave velocities in the Opalinus Clay than for the geomechanical rock parameters, which we discussed in the manuscript (page 23, lines 465-467 in the marked manuscript):

*"As ultrasonic velocity is explicitly dependent on the internal structure of the rock, such as cementation, anisotropy and porosity structure (Jaeggi and Bossart, 2014; Schuster et al., 2017), it can be assumed that the relationship of ultrasonic velocity and NPI is probably rather weak."*

However, since these are published equations, we find it important to show and evaluate the results for the estimated geophysical parameters in addition to the geomechanical parameters.

Reviewer 2 criticises the statistical basis of the conclusions your reach about the EDZ fractures because your analyzed dataset is very small. You don't acknowledge this small dataset well enough in the current conclusions. Please add a statement about this.

**Response**: We agree that the analyzed EDZ fracture dataset is rather small due to the limited number of accessible measurement points in the niche, but we are fully convinced that the data and the conclusion is still important for the reader. However, we agree that this point should also be addressed in the conclusion and we have therefore revised the respective text passage and added a statement as requested (page 24, lines 513-514 in the marked manuscript):

*"Due to the smaller mean mechanical aperture of the artificially induced unloading fractures compared to the investigated tectonic fractures, conversion was most appropriate for the EDZ fractures. Tectonic fractures on average exhibit a higher variance of measured distances along imaged fracture traces, which can be explained by a higher degree of mismatch between the fracture surfaces due to the reactivation of fault planes during excavation. However, the statistical significance of the observed differences between the different fracture types would have to be tested based on a larger dataset."*

In addition, we have already included a statement on the small size of the EDZ fracture data set in Section 3.1.3, as stated in our response to comments of reviewer #2 (page 18, lines 374-376 in the marked manuscript):

*"While EDZ fractures seem to better correspond to the general model concept of an "ideal plane-parallel fracture", tectonic fractures are most probably characterized by a different $a_m$–$a_h$–relation. While implied by the presented measurement data, this issue should still be examined based on larger data sets."*

THMC at line 47. The reviewer's point was that you never define the full term, just go straight to the acronym. You must give the full version here too.

**Response**: We agree and therefore inserted the full term in addition to the acronym at this point (page 2, line 47 in the marked manuscript):

*"[…] and to examine the behavior of the Opalinus Clay when exposed to short- or long-term THMC (thermal, hydrological, mechanical and chemical) impacts (Bossart et al., 2017; Pearson et al., 2003)."*

Line 142: I don't find your revision clear enough. Why not just clearly say 'within the first 1.3 m depth into the niche'?

**Response**: We agree and have changed the corresponding sentence in the manuscript as suggested (page 5, line 145 in the marked manuscript):

*"Excavation-induced unloading joints (EDZ fractures) that are related to the construction of Gallery 04 are present within the first 1.3 m depth into the EZ-B niche (Nussbaum et al., 2005)."*